# Efficient Lifelong Model Evaluation in an Era of Rapid Progress

**Ameya Prabhu**[*1,3]    **Vishaal Udandarao**[*1,2]    **Philip H.S. Torr**[3]
**Matthias Bethge**[1†]    **Adel Bibi**[3†]    **Samuel Albanie**[2†]

[1]Tübingen AI Center, University of Tübingen    [2]University of Cambridge    [3]University of Oxford

 https://github.com/bethgelab/sort-and-search
 https://huggingface.co/datasets/bethgelab/lifelong_benchmarks

## Abstract

Standardized benchmarks drive progress in machine learning. However, with repeated testing, the risk of overfitting grows as algorithms over-exploit benchmark idiosyncrasies. In our work, we seek to mitigate this challenge by compiling *ever-expanding* large-scale benchmarks called *Lifelong Benchmarks*. These benchmarks introduce a major challenge: the high cost of evaluating a growing number of models across very large sample sets. To address this challenge, we introduce an efficient framework for model evaluation, *Sort & Search (S&S)*, which reuses previously evaluated models by leveraging dynamic programming algorithms to selectively rank and sub-select test samples. To test our approach at scale, we create *Lifelong-CIFAR10* and *Lifelong-ImageNet*, containing 1.69M and 1.98M test samples for classification. Extensive empirical evaluations across ∼31,000 models demonstrate that *S&S* achieves highly-efficient approximate accuracy measurement, reducing compute cost from 180 GPU days to 5 GPU hours (∼1000x reduction) on a single A100 GPU, with low approximation error and memory cost of <100MB. Our work also highlights issues with current accuracy prediction metrics, suggesting a need to move towards sample-level evaluation metrics. We hope to guide future research by showing our method's bottleneck lies primarily in generalizing *Sort* beyond a single rank order and not in improving *Search*.

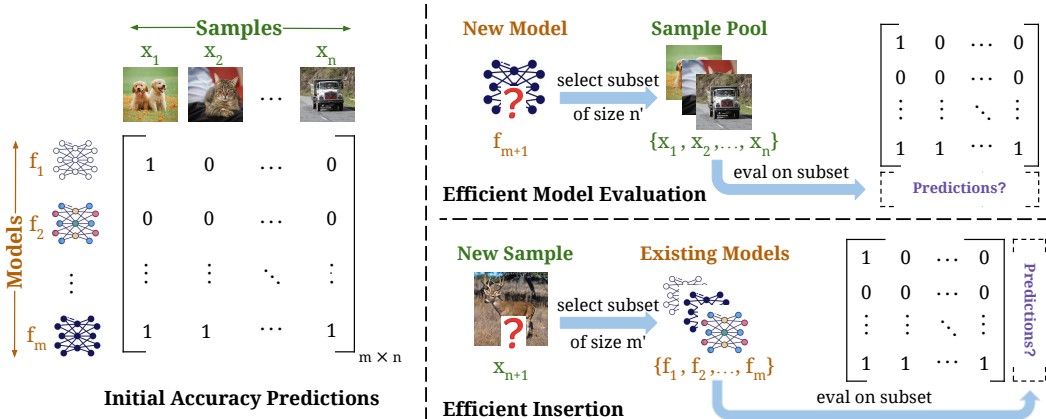

Figure 1: *Efficient Lifelong Model Evaluation.* Assume an initial pool of $n$ samples and $m$ models evaluated on these samples *(left)*. Our goal is to efficiently evaluate a new model (insert$_\mathcal{M}$) at sub-linear cost *(right top)* and efficiently insert a new sample into the lifelong benchmark (insert$_\mathcal{D}$) by determining sample difficulty at sub-linear cost *(right bottom)*. See Section 2 for more details.

---

[*]equal contribution, † equal supervising

38th Conference on Neural Information Processing Systems (NeurIPS 2024).

# 1  Introduction

The primary goal of standard evaluation benchmarks is to assess model performance on some task using data that is *representative of the visual world* [87]. For instance, the CIFAR10 [54] benchmark tested whether classifiers can distinguish between 10 categories, such as dogs and cats. Subsequent versions like CIFAR10.1 [59], CIFAR10.2 [59], CINIC10 [21], and CIFAR10-W [83] introduced more challenging and diverse samples to evaluate the *same objective of classifying 10 categories*. As benchmarks become standardized and repeatedly used to evaluate competing methods, they gradually lose their capacity to represent broader tasks effectively. This is because models become increasingly specialized to perform well on these specific benchmarks. This phenomenon, known as overfitting, occurs both in individual models and within the research community as a whole [28, 90]. Fresh approaches must compete with a body of methods that have been highly tuned to such benchmarks, incentivising further overfitting if they are to compete [9, 10].

One approach to preventing models from overfitting to biases [87, 3] is to move beyond fixed test sets by creating an ever-expanding pool of test samples. This approach, known as *Lifelong Model Evaluation*, aims to restore the representativeness of benchmarks to reflect the diversity of the visual world by expanding the coverage of test sets. One can expand the pool by combining datasets or using well-studied techniques like dynamic sampling [81, 51, 52], these expanding benchmarks can grow substantially in size as they accumulate samples. This raises the less-explored issue of increasing evaluation costs. As an example, it takes roughly 140 and 40 GPU days respectively to evaluate our current model set on our *Lifelong-CIFAR10* and *Lifelong-ImageNet* datasets (containing 31,000 and 167 models respectively). These issues are only exacerbated in benchmarking foundation models [15]. For instance, evaluating a single large language model (LLM) on MMLU [40] (standard benchmark for evaluating LLMs) takes 24 hours on a consumer-grade GPU [45]. This inevitably will lead to a surge in evaluation costs when benchmarking lots of increasingly expensive models against an ever-growing collection of test samples [78, 22]. Hence, we primarily ask: *Can we reduce this evaluation cost while minimising the prediction error?*

We design algorithms to enable efficient evaluation in lifelong benchmarks, inspired by computerized adaptive testing (CAT) [89]. CAT is a method used to create exams like the GRE and SAT from a continuously growing pool of questions. Unlike traditional tests where all questions must be answered, CAT sub-samples questions based on examinee responses. This approach efficiently gauges proficiency with far fewer questions, while maintaining assessment accuracy. Similarly, we aim to evaluate classification ability of new models without testing on all samples, instead selecting a subset of samples to evaluate models. We propose a method, *Sort & Search (S&S)*, which reuses past model evaluations on a sample set through dynamic programming to enable efficient evaluation of new incoming models. *S&S* operates by first ranking test samples by their difficulty, done efficiently by leveraging data from previous tests. It then uses these updated rankings to evaluate new models, streamlining the benchmarking process. This strategy enables efficient lifelong benchmarking, reducing the cost dramatically from a collective of 180 GPU days to 5 GPU hours on a single A100 GPU. We achieve a $1000\times$ reduction in inference costs compared to static evaluation on all samples, reducing over 99.9% of computation costs while accurately predicting sample-wise performance. Moreover, with a single algorithm, we address both key challenges: expanding dataset size and evaluating new models given a dataset.

Taken together, our main contributions are:

1. We curate two lifelong benchmarks: *Lifelong-CIFAR10* and *Lifelong-ImageNet*, consisting of 1.69M and 1.98M samples respectively.
2. We propose *Sort & Search*, a novel framework for efficient model evaluation.
3. We show that our simple framework is far more scalable and allows saving 1000x evaluation cost.
4. We provide a novel decomposition of errors in *Sort & Search* into largely independent sub-components (aleatoric and epistemic errors).
5. We prove and empirically validate that our solution for the *Search* sub-component reaches the optimal solution and our framework is stable under repeated additions without any degradation.

# 2  Lifelong Model Evaluation: Formulation and Challenges

We first formalise evaluation in lifelong model evaluation and describe the key challenges it raises.

**Formulation.** Let $\mathcal{D}=((x_1,y_1),\ldots,(x_n,y_n))$ denote an ordered collection of labeled examples, sampled from the underlying task distribution of interest $P(\mathcal{X}\times\mathcal{Y})$. Here, $x_i\in\mathcal{X}$ denotes the $i^{\text{th}}$ data sample and $y_i\in\mathcal{Y}$ denotes the corresponding label. Let $\mathcal{M}=(f_1,\ldots,f_m)$ denote an ordered collection of models where each model, $f:\mathcal{X}\to\mathcal{Y}$, maps data samples to predicted labels. *Lifelong benchmark*, $\mathcal{B}=(\mathcal{D},\mathcal{M},\texttt{insert}_{\mathcal{D}},\texttt{insert}_{\mathcal{M}},\texttt{metrics})$, augments $\mathcal{D}$ and $\mathcal{M}$ with three operations:

**1** $\texttt{insert}_{\mathcal{D}}((x',y'))$ inserts a new labeled example $(x',y')$ into $\mathcal{D}$.

**2** $\texttt{insert}_{\mathcal{M}}(f')$ inserts a new model $f'$ into $\mathcal{M}$.

**3** $\texttt{metrics}()$ returns a $|\mathcal{M}|$-dimensional vector estimating each model's performance.

**Key challenges.** When new models are proposed, the set $\mathcal{M}$ expands over time. Similarly, the sample collection, $\mathcal{D}$ expands as new evaluation datasets get proposed to test various aspects of the problem and resist overfitting. The key question becomes: How to efficiently update the benchmark? We can instantiate a "naive" implementation of the $\texttt{metrics}()$ operation (**3**) by simply re-evaluating every model on every sample after each call to $\texttt{insert}_{\mathcal{M}}$ (**2**) or $\texttt{insert}_{\mathcal{D}}$ (**1**). However, such a strategy exhibits $O(|\mathcal{D}||\mathcal{M}|)$ runtime complexity for each call to $\texttt{metrics}()$, rendering lifelong model evaluation practically infeasible as $\mathcal{D}$ and $\mathcal{M}$ grow. The central question considered by this work is therefore the following: *Given a lifelong benchmark $\mathcal{B}$, how can we efficiently compute* $\texttt{metrics}()$ *each time we insert new labeled samples into $\mathcal{D}$ (**1**) or new models into $\mathcal{M}$ (**2**)?*

**Inserting $\Delta m$ models (**2** $\texttt{insert}_{\mathcal{M}}$).** Suppose that $\Delta m$ new models have just been released. We wish to insert these new models into $\mathcal{M}$ and efficiently predict performance of these new models. A naive approach would entail evaluating the $\Delta m$ models on all $|\mathcal{D}|$ samples. Our first challenge is: Can we instead generate the prediction matrix by performing inference only on a small subset of $n'\ll|\mathcal{D}|$ samples? We want to enable accurate prediction of the remaining entries in the prediction matrix.

**Inserting $\Delta n$ samples (**1** $\texttt{insert}_{\mathcal{D}}$).** Our second challenge arises when we obtain new $\Delta n$ labeled data examples. We seek to insert these samples into $\mathcal{D}$ and efficiently predict performance of these new samples. A naive approach entails evaluating all $|\mathcal{M}|$ models on the $\Delta n$ new examples. As above, to substantially reduce cost, we select a small subset of $m'\ll|\mathcal{M}|$ models with the objective of accurately predicting the remaining entries of the prediction matrix corresponding to the new $\Delta n$ samples.

**Approach.** Our approach is characterized by two key ideas. First, we augment $\mathcal{B}$ with an *instance-level accuracy cache* to amortise inference costs across evaluations. The cache is instantiated as a matrix $\mathbf{A} \in \{0,1\}^{|\mathcal{M}|\times|\mathcal{D}|}$ where $\mathbf{A}(i,j) \triangleq \mathbb{I}[f_i(x_j) = y_j]$. Second, we propose strategies to efficiently generate the prediction matrix $\mathbf{Y} \in \{0,1\}^{|\mathcal{M}|\times|\mathcal{D}|}$, using a combination of sampling and inference leveraging the accuracy cache. Our methodology is illustrated in Fig. 1.

**Connections to Existing Literature.** The lifelong model evaluation setup, where $\mathcal{M}$ and $\mathcal{D}$ grow over time, has received limited attention [3], the sub-challenge of efficiently evaluating models when new models are released has received more focus. Concretely, this maps to the problem of $\texttt{insert}_{\mathcal{M}}$ (**2**) within our framework. We comprehensively draw connections across different research directions in Appendix H and briefly present the most similar works here. Model Spider [105] efficiently ranks models from a pre-trained model zoo. LOVM [110], Flash-Eval [106] and Flash-HELM [67] similarly rank foundation models efficiently on unseen datasets. However, these approaches predict dataset-level metrics rather than instance-level metrics, and thereby cannot be used in our setup to grow the prediction cache efficiently (see Section 2.1). Concurrent to our work, Anchor Point Sampling [91] and IRT-Clustering [69] both propose efficient instance-level evaluations by creating smaller core-sets from test data. They introduce clustering-based approaches and item response theory [4] to obtain sample-wise accuracy predictions. However, their methods require memory and time complexity quadratic in the number of data samples, i.e., $\mathcal{O}(|\mathcal{D}|^2)$ requiring well over 10TB of RAM for benchmarks having a million samples. The comparisons are infeasible to scale on datasets bigger than a few thousand samples. In contrast, our novel *Sort & Search* approach, requires memory and time complexity of $\mathcal{O}(|\mathcal{D}|\log|\mathcal{D}|)$ with the number of samples, and can scale up to billion-sized test sets (see Section 4 for empirical results). In practice, our method only requiring only two 1D arrays of size of the number of samples, requiring extremely minimal storage overhead, being less than 3GB in absolute terms on billion scale datasets. Furthermore, we motivate why one should adopt sample-wise prediction instead of overall accuracy prediction below.

## 2.1 Why Adopt Sample-wise Prediction Metrics instead of Overall Accuracy Prediction?

Given model predictions $\mathbf{y}_{m+1}$ and ground-truth predictions $\mathbf{a}_{m+1}$, current methods typically measures whether one can predict the average accuracy over the full test, measured by mean absolute difference of aggregate accuracies $E_{\text{agg}}(\mathbf{y}_{m+1}, \mathbf{a}_{m+1}) = |(|\mathbf{y}_{m+1}| - |\mathbf{a}_{m+1}|)|/n$. We argue this is highly unreliable as minimizing the metric only requires predicting the count of 1s in the prediction array rather than correctly predicting on a sample level. For instance, consider a ground-truth prediction array of [0,0,0,1,1,1]. A method that predicts [1,1,1,0,0,0] as the estimated prediction array achieves optimal $E_{\text{agg}}$ of 0 despite not predicting even a single sample prediction correctly! More generally, it is always possible to obtain globally optimal $E_{\text{agg}}$ of 0 while having worst-case mean-absolute error $E$ for any ground truth accuracy $a_{m+1}$. Formally,

**Theorem 2.1.** *Given any ground-truth vector* $\mathbf{a}_{m+1}$, *it is possible to construct a prediction vector* $\mathbf{y}_{m+1}$ *such that* $E_{agg}(\mathbf{y}_{m+1}, \mathbf{a}_{m+1}) = 0$ *and* $E(\mathbf{a}_{m+1}, \mathbf{y}_{m+1}) = 2.min(1 - |\mathbf{a}_{m+1}|/n, |\mathbf{a}_{m+1}|/n)$

One might wonder whether these worst case bounds ever occur in practice. We empirically test a simple yet optimal array construction, given with oracle ground-truth dataset-level accuracy of $k^2$, which achieves $E_{\text{agg}} = 0$, and consistently observe high mean-absolute error $E$ of $0.4-0.5$ on a sample level on our lifelong benchmarks ($n=\sim 10^6$), *i.e.*, the model incorrectly predicts $40-50\%$ of the samples in a binary classification task, which is surprisingly high. In comparison, our *S&S* method, without any oracle access, gets $0.15-0.17$ mean-absolute error with just $n'=100$ samples (at $10,000$x compute saving) on the same benchmarks. Overall, this demonstrates that thoughtful sample-level prediction mechanisms are necessary for efficient lifelong evaluation.

## 3 *Sort & Search*: Enabling Efficient Lifelong Model Evaluation

Inspired by CAT [89], we propose an efficient lifelong evaluation framework, *Sort & Search (S&S)*, comprising two components: (1) Ranking test samples from the entire dataset pool according to their difficulty[3], *i.e.*, *Sort* and (2) Sampling a subset from the pool to test on, *i.e.*, *Search*. This framework effectively tackles the two key operations noted in Section 2 (**1** insert$_{\mathcal{D}}$ and **2** insert$_{\mathcal{M}}$).

We first describe our Sort and Search method in the case when new models are added (**2** insert$_{\mathcal{M}}$), and subsequently show that the same procedure applies when we have new incoming samples (**1** insert$_{\mathcal{D}}$) simply by transposing the cache ($\mathbf{A} \rightarrow \mathbf{A}^T$). A full schematic of our pipeline is depicted in Fig. 2.

### 3.1 Ranking by Sort

**Setup.** We recall that our lifelong benchmark pool consists of evaluations of $|\mathcal{M}|$ models on $|\mathcal{D}|$ samples. For ease of reference, say $|\mathcal{M}|=m$ and $|\mathcal{D}|=n$, and we have our cache $\mathbf{A} \in \{0,1\}^{m \times n}$ (see Fig. 1 left). We can decompose the cache $\mathbf{A}$ row-wise corresponding to each model $f_i$, $i \in \{1,..,m\}$, obtaining the binary accuracy prediction across the $n$ samples, denoted by $\mathbf{a}_i = [p_{i1}, p_{i2} \ldots, p_{in}]$. Here, $p_{ij} \in \{0, 1\}$ represents whether the model $f_i$ classified the sample $x_j$ correctly.

**Goal.** Given the cache $\mathbf{A}$, we want to obtain a ranked order (from easy to hard) for its columns, which represent the samples. This sorted order (*Sort*) can later be used for efficient prediction on new incoming models (*Search*). We want to find the best global permutation matrix $\mathbf{P} \in \{0,1\}^{n \times n}$, a binary matrix, such that $\mathbf{AP}$ permutes the *columns* of $\mathbf{A}$ so that we can rank *samples* from *easy* (all 1s across models) to *hard* (all 0s across all models). We say this has a minimum distance from the optimal ranked accuracy prediction matrix $\mathbf{Y} \in \{0,1\}^{m \times n}$ computed by the hamming distance between them, posed as solving the following problem:

$$\mathbf{P}^*, \mathbf{Y}^* = \text{argmin}_{\mathbf{P},\mathbf{Y}} \|\mathbf{AP} - \mathbf{Y}\|_1, \qquad s.t. \quad \mathbf{P} \in \{0,1\}^{n \times n}, \mathbf{P}\mathbf{1}_n = \mathbf{1}_n, \mathbf{1}_n^\top \mathbf{P} = \mathbf{1}_n,$$
$$\text{if} \quad \mathbf{Y}_{ij} = 1, \text{then } \mathbf{Y}_{ij'} = 1 \ \forall j' \leq j, \qquad \text{if} \quad \mathbf{Y}_{ij} = 0, \text{then } \mathbf{Y}_{ij'} = 0 \ \forall j' \geq j. \tag{1}$$

---

[2]Given $|\mathbf{a}_{m+1}| = k$, one can show the prediction array $[\mathbf{1}_k^\top, \mathbf{0}_{n-k}^\top]$ achieves optimal $E_{agg} = 0$

[3]If a sample $x_i$ is more "difficult" than a sample $x_j$ then at least equal number of models predict $x_j$ correctly as the number of models predicting $x_i$ correctly [6].

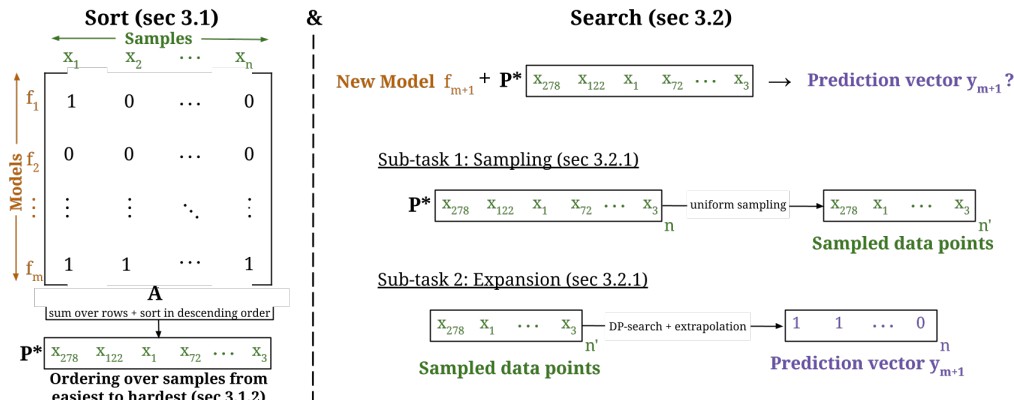

Figure 2: **Full Pipeline of *Sort & Search*.** For efficiently evaluating new models, *(Left)* we first sort all data samples by difficulty (refer Section 3.1) and *(Right)* then perform a uniform sampling followed by DP-search and extrapolation for yielding new model predictions (refer Section 3.2). This entire framework can also be transposed to efficiently insert new samples (refer Section 3.3).

By definition of a permutation matrix, the constraints $\mathbf{P}\mathbf{1}_n = \mathbf{1}_n, \mathbf{1}_n^\top\mathbf{P} = \mathbf{1}_n$ on binary $\mathbf{P}$ enforces by definition that $\mathbf{P}$ is a valid permutation matrix. The ranked accuracy prediction matrix $\mathbf{Y}$ is a binary matrix created by a row-wise application of a thresholding operator for every row in $\mathbf{Y}$ separately. The informal explanation of the optimization problem in Eq. (1) is to find an ordering of samples such that error introduced by thresholding is minimized.

We next discuss how to solve this optimization. While the goal is finding the optimal permutation $\mathbf{P}^*$, we still need to jointly solve for $\mathbf{P}, \mathbf{Y}$ here. We find a solution by alternating between optimizing $\mathbf{P}$ keeping $\mathbf{Y}$ constant and optimizing $\mathbf{Y}$ keeping $\mathbf{P}$ constant, with the goal of finding the best $\mathbf{P}^*$, with a coordinate descent algorithm. We now present algorithms for optimizing the two subproblems.

### 3.1.1 Optimizing P Given Y

We know $\mathbf{P}$ is binary from Eq. (1). Hence, finding the optimal $\mathbf{P}^*$ is NP-Hard [101]. To simplify the sub-problem, we first present an algorithm to solve the case where we can order samples in a strictly decreasing order of difficulty, measured by how many models classified it correctly (❶). However, samples cannot be arranged as strictly decreasing in practice. Subsequently, we present an alternative which computes soft confidences, enabling the strictly decreasing constraint to hold (❷). A third alternative we explore removes the introduced constraint of a strictly decreasing order (❸).

❶ **Sorting by Sum.** We discuss how to order samples if they follow a strictly decreasing order of difficulty. We can order samples in decreasing order of difficulty by a simple algorithm detailed in Listing 1 (`sort_by_sum`)—intuitively, this algorithm greedily sorts samples from easy (more 1s) to hard (less 1s) by sorting the sum vector across rows per column (which can trivially be converted to the permutation matrix $\mathbf{P}^*$).

However, the assumption of strictly decreasing order of difficulty is unrealistic as the number of samples is usually far larger than the number of models. Hence, it is guaranteed that many samples will have the same level of difficulty by the pigeonhole principle [2]. We propose to address this by two methods: (a) Converting the cache (A) to store confidence predictions of ground truth class rather than accuracy (Algorithm ❷), or (b) Iteratively optimizing rows which are tied in sum values (Algorithm ❸). Note that we find ❶ Sorting by Sum effective in all our tested scenarios, but provide these alternatives in the case where it is insufficient.

❷ **Sorting by Confidence Sum.** One method to have a strictly decreasing order is to relax the constraint on the samples of $\mathbf{a}_i = [p_{i1}, p_{i2} \ldots, p_{in}]$ from $p_{ij} \in \{0, 1\}$ to $p_{ij} \in [0, 1]$, and use confidence of the ground truth class. This modification allows all examples to be unique. The procedure is then identical to Sorting by Sum, i.e. algorithm still greedily sorts samples from easy (more 1s) to hard (less 1s) by sorting the sum vector across rows per column.

❸ **Recursive Sorting by Sum.** Another alternative is relaxing the equal difficulty assumption in Algorithm ❶. A natural question is: *How does one order samples which have equal number of models predicting them correctly, i.e., two columns of* $\mathbf{A}$ *with equal number of* 1s?

We propose an iterative solution: at each step, order samples of equal difficulty by alternatively optimizing $\mathbf{P}$ keeping $\mathbf{Y}$ constant by applying Algorithm ❶ and optimizing $\mathbf{Y}$ keeping $\mathbf{P}$ constant by *DP-Search* algorithm (presented in the next Section). We provide the algorithm for two iterations for an illustration in Listing 1 (`two_stage_sort_by_sum`). Note that this strictly improves the solution at each recursion depth. Note that ties are broken by preferring the model which minimizes error the most.

### 3.1.2 Optimizing Y given a P

Optimizing $\mathbf{Y}$ given a $\mathbf{P}$, is equivalent to finding a row-wise threshold $k \leq n$ minimizing the error with the matrix $\mathbf{AP}$ for a given $\mathbf{P}$. Intuitively, if the threshold for the $i^{\text{th}}$ row is $k$, then the $i^{\text{th}}$ row is of the form $[\mathbf{1}_k^\top, \mathbf{0}_{n-k}^\top]$ where $\mathbf{1}_k$ is a vector of all ones of size $k$ and $\mathbf{0}_{n-k}$ is a zero vector of size $n-k$. In every row, all samples before the row-wise threshold $k$ are predicted to be correctly classified (easy) and those after are incorrectly classified (hard) for the model corresponding to the row. To optimize $\mathbf{Y}$ given $\mathbf{P}$, we propose a dynamic programming algorithm, *DP-Search* which operates on each row $\mathbf{y}_i$, detailed in Listing 1 (`dp_search`). Given a row in $\mathbf{Y}$, DP-Search computes the difference between number of 1s and number of 0s for each index. By using a prefix sum structure, for an input of size $n$, the DP approach reduces time complexity from $\mathcal{O}(n^2)$ to $\mathcal{O}(n)$. The optimal threshold $k$ is the index of the maximum value in this vector. The vector $\mathbf{y}_i$ is simply $[\mathbf{1}_k^\top, \mathbf{0}_{n-k}^\top]$ where $\mathbf{1}_k$ is a vector of all ones of size $k$ and $\mathbf{0}_{n-k}$ is a zero vector of size $n-k$. *DP-Search* is guaranteed to return the globally optimal solution:

**Theorem 3.1.** *Optimality of* $\mathbf{Y}$ *given* $\mathbf{P}$. *For any given* $\mathbf{a}_i \in \{0,1\}^{1 \times n}$ *and* $\mathbf{P}$, *DP-Search returns an ordered prediction vector* $\mathbf{y}_i \in \{0,1\}^{1 \times n}$ *which is a global minimum of* $\|\mathbf{a}_i\mathbf{P} - \mathbf{y}_i\|_1$.

Applying *DP-Search* independently row-wise, the algorithm returns the optimal $\mathbf{Y}$ given $\mathbf{P}$. Now, we shall

### 3.1.3 Process Summary

We have outlined the process of optimizing (i) $\mathbf{P}$ given $\mathbf{Y}$ and (ii) $\mathbf{Y}$ given $\mathbf{P}$. Note that (i) alone suffices for Sorting Operation when using the ❶ Sorting by Sum algorithm, while a combination of (i) and (ii) is primarily needed for ❸ Recursive Sorting by Sum. After sorting, we obtain $\mathbf{AP}^*$, which reflects the sample ordering based on difficulty. In the following section, we will reuse (ii) to search for a $\mathbf{Y}$ given $\mathbf{P}$ to efficiently evaluate new models or add new samples.

## 3.2 Efficient Selection by Search

**Goal.** After solving Eq. (1), we obtain the optimal $\mathbf{P}^*$ in the sorting phase. We assume that this sample difficulty order generalizes to new models, $\Delta m$. Recall that $\mathbf{AP}^*$ represents the columns of cache A, ordered by sample difficulty (those most often misclassified by models). Given $\Delta m$ new models, our goal is to predict accuracy across all $n$ samples for each model, i.e., the accuracy matrix $\mathbf{Y}_{\Delta m} \in \{0,1\}^{\Delta m \times n}$. This would be simple if we could evaluate all $\Delta m$ models on all $n$ samples, but this approach is costly. The challenge is thus to predict performance on the remaining samples while evaluating only a small subset $n' \ll n$. Hence, we will assume that we can create a smaller ground truth subset $\mathbf{a}'_{m+1}$ and study: How to find the best accuracy prediction vector $\mathbf{y}_{m+1}$? We use the ground truth vector $\mathbf{a}_{m+1}$ for evaluating the efficacy of our method.

Recall that evaluation of every new model can be done independently of others, i.e. $\mathbf{Y}_{\Delta m}$ is separable per row. Hence, we describe the problem for the first new model $\mathbf{y}_{m+1} \in \{0,1\}^{1 \times n}$ here.

**(i) How to get the optimal** $\mathbf{y}_{m+1}$**?** Our goal here is to generate the sample-wise prediction vector $\mathbf{y}_{m+1} \in \{0,1\}^{1 \times n}$. We divide it into two subtasks: *selection* and *optimization*. The selection task is to select the best $n'$ observations to sample. The optimization task is, given the $n'$ observations $\mathbf{a}'_{m+1} \in \{0,1\}^{1 \times n'}$ how to generate the prediction vector $\mathbf{y}_{m+1} \in \{0,1\}^{1 \times n}$.

*Subtask 1: How to Select Samples?* We want to find the best $n'$ observations forming $\mathbf{a}'$. Note that any ranked solution we obtain using this vector needs to be interpolated from $n'$ points to $n$ points— we use this intuition to sample $n'$ points. Hence, a simple solution is to sample points such that any threshold found minimizes the difference between the actual threshold and a threshold predicted by our set of $n'$, *i.e.*, sample $n'$ points uniformly, providing the algorithm in Listing 1 (`uniform_sampling`). We also compare empirically with a pure random sampling approach in Section 4.

*Subtask 2: Optimizing* $\mathbf{y}_{m+1}$. Given the $n'$ observations $\mathbf{a}'_{m+1} \in \{0,1\}^{1 \times n'}$, how to generate the prediction vector $\mathbf{y}_{m+1} \in \{0,1\}^{1 \times n}$? We use the threshold given by *DP-Search* (Listing 1) and obtain the threshold, given in terms of fraction of samples in $|\mathbf{a}'_{m+1}|$. We extrapolate this threshold from $n'$ to $n$ points, to obtain the threshold for the prediction vector $\mathbf{y}_{m+1}$. $\mathbf{y}_{m+1}$ is simply $[\mathbf{1}_k^\top, \mathbf{0}_{n-k}^\top]$ where $\mathbf{1}_k$ is a vector of all ones of size $k$ and $\mathbf{0}_{n-k}$ is a zero vector of size $n - k$.

So far, we have only discussed evaluation of $\Delta m$ new models (❷ insert$_{\mathcal{M}}$). How can we also efficiently extend the benchmark *i.e.* efficiently adding $\Delta n$ new samples (❶ insert$_{\mathcal{D}}$)?

### 3.3   Efficient Insertion of New Samples (insert$_{\mathcal{D}}$)

To add new samples into our lifelong benchmark efficiently, we have to estimate their difficulty with respect to the other samples in the cache $\mathbf{A}$. To efficiently determine difficulty by only evaluating $m' \ll m$ models, a ranking over models is required to enable optimally sub-sampling a subset of $m'$ models. This problem is quite similar in structure to the previously discussed addition of new models, where we had to evaluate using a subset of $n' \ll n$ samples. *How do we connect the two problems?*

We recast the same optimization objectives as described in Eq. (1), but replace $\mathbf{A}$ with $\mathbf{A}^\top$ and $\mathbf{Y}$ with $\mathbf{Y}^\top$. In this case, Eq. (1) would have $\mathbf{A}^\top \mathbf{P}$, which would sort models, instead of samples, based on their aggregate sum over samples (*i.e.*, accuracy) optimized using Algorithm ❶ to obtain $\mathbf{P}^*$, ordering the models from classifying least samples correctly to most samples correctly. Here, Algorithm ❶ is sufficient, without needing to solve the joint optimization (❸) because accuracies (sum across rows) are unique as the number of samples is typically much larger than the number of models. In case of new incoming samples $\Delta n$, we similarly would treat every sample independently and optimize the predicted array $\mathbf{y}_{n+1}^\top$ using *Efficient Selection by Search* (Section 3.2).

## 4   Experiments

To validate *Sort & Search* empirically, we showcase experiments on two tasks: ❶ *efficient estimation of new sample difficulties* (insert$_{\mathcal{D}}$) and ❷ *efficient performance evaluation of new models* (insert$_{\mathcal{M}}$). We then comprehensively analyse various design choices within our *S&S* framework.

### 4.1   Experimental Details

**Lifelong-Datasets.** We combine 31 domains of different CIFAR10-like datasets comprising samples with various distribution shifts, synthetic samples generated by diffusion models, and samples queried from different search engines to form *Lifelong-CIFAR10*. We deduplicate our dataset and downsample images to 32×32. Our final dataset consists of 1.69M samples. Similarly, we source test samples from ImageNet and corresponding variants to form *Lifelong-Imagenet*, designed for increased sample diversity (43 unique domains) while operating on the same ImageNet classes. We include samples from different web-engines and generated using diffusion models. Our final *Lifelong-ImageNet* contains 1.98M samples (see full list of dataset breakdown in Appendix C).

**Model Space.** For *Lifelong-CIFAR10*, we use $31,250$ CIFAR-10 pre-trained models from the NATS-Bench-Topology-search space [25]. For *Lifelong-ImageNet*, we use 167 ImageNet-1K and ImageNet-21K pre-trained models, sourced primarily from `timm` [98] and `imagenet-testbed` [84].

**Sample Addition Split** (❶ insert$_{\mathcal{D}}$)**.** To study efficient estimation of new sample difficulties on *Lifelong-CIFAR10*, we hold-out CIFAR-10W [83] samples for evaluation ($\sim 500,000$ samples) and use the rest $\sim 1.2$ million samples for sorting. We do not perform experiments for *Lifelong-Imagenet* since the number of models is quite small (167 in total), directly evaluating all models is relatively efficient, as opposed to the more challenging *Lifelong-CIFAR10* where evaluation on $31,250$ models is expensive, practically necessitating reducing the number of models evaluated per new sample.

**Model Evaluation Split** (❷ insert$_{\mathcal{M}}$)**.** To study efficient evaluation of new models, we split the model set for the *Lifelong-CIFAR10* benchmark into a randomly selected subset of $6,000$ models for ordering samples (*i.e., Sort*) and evaluate metrics on the remaining $25,250$ models (*i.e., Search*). For *Lifelong-Imagenet*, we use 50 random models for ordering samples and evaluate on 117 models.

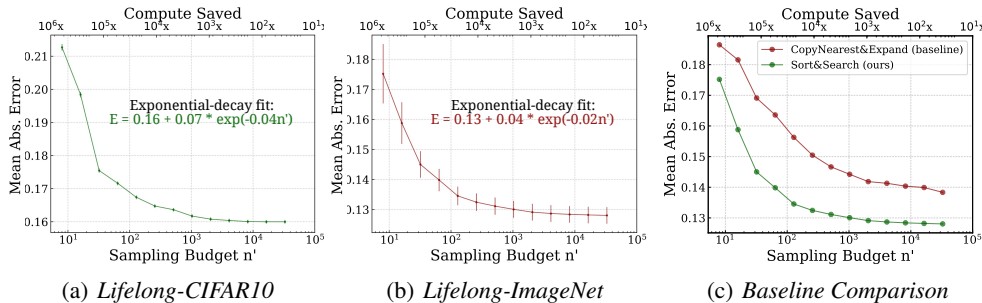

(a) *Lifelong-CIFAR10*  (b) *Lifelong-ImageNet*  (c) *Baseline Comparison*

Figure 3: **Main Results.** *(a,b)* We achieve 99% cost-savings for new model evaluation on *Lifelong-ImageNet* and *Lifelong-CIFAR10* showcasing the efficiency (MAE decays exponentially with $n'$) of *Sort&Search*. *(c)* S&S is more efficient and accurate compared to the baseline on *Lifelong-ImageNet*.

**Metrics (③ metrics()).** We measure errors between estimated predictions for each new model $\mathbf{y}_{m+1}$ and ground-truth predictions $\mathbf{a}_{m+1}$ using mean-absolute error (MAE): $E(\mathbf{a}_{m+1}, \mathbf{y}_{m+1})$:

$$E(\mathbf{a}_{m+1}, \mathbf{y}_{m+1}) = \|\mathbf{a}_{m+1}\mathbf{P}^* - \mathbf{y}_{m+1}\|_1 / n \qquad (2)$$

## 4.2 Results: Sample-Level Model Performance Estimation (insert$_\mathcal{M}$)

We evaluate the predictive power of *S&S* for evaluating new models (②) when subjected to a varying sampling budgets $n'$ *i.e.,* we run our *S&S* over 13 different sampling budgets: {8, 16, 32, 64, 128, 256, 512, 1024, 2048, 4096, 8192, 16384, 32768} on both *Lifelong-ImageNet* and *Lifelong-CIFAR10*.

Our main results in Sections 4.2 and 4.3 use *Sorting by Sum* (①) for obtaining $\mathbf{P}^*$ and uniform sampling for the sample budget $n'$. Using this configuration, we now present our main results.

**Key Result 1: Extreme Cost-Efficiency.** From Figs. 3(a) and 3(b), we note our approach converges to a very low mean-absolute error with $1/1000$ the number of evaluation samples, leading to extreme cost savings at inference time (from 180 GPU days to 5 GPU hours on one A100-80GB GPU)[4].

**Key Result 2: Mean Absolute Error Decays Exponentially.** Upon analysing the observed $E$ vs. $n'$ relationship, we note that exponentially decreasing curves fit perfectly in Figs. 3(a) and 3(b). The exponential decay takes the form $E = ae^{-bx} + c$. The fitted curves have large exponential coefficients $b$ of 0.04 and 0.02. This further shows the surprisingly high sample-efficiency obtained by *S&S*.

**Key Result 3: Outperforming Baselines by Large Margins.** We construct a competitive, scalable version of Vivek et al. [91] as a baseline, called *CopyNearest&Expand*: It first samples $n'$ points out of $n$ (similar to *S&S* without sorting), and then expands the $n'$-sized prediction array to $n$ samples by copying the rest $n-n'$ predictions from the nearest neighbor prediction array from the ranking set of models. We note that this baseline is equivalent to removing the *Sort* component, and only using random sampling. Comparing to the baseline, we see from Fig. 3(c) that our *Sort & Search* is:

*1) More accurate:* It achieved 1% lower MAE at a sampling budget of $n'$=8192 compared to the baseline, meaning that on average, our *S&S* correctly classifies ∼19k more samples.

*2) Faster convergence:* *S&S* converges much faster than the baseline (at $n'$=1,024 vs. $n'$=32,768) thereby showcasing the high degree of sample efficiency in converging to the minimal error.

*3) Consistent:* Fig. 4(b) shows the better consistency of *S&S*, across wider range of models used for *Sort*—at $n'$=512, *S&S* with only 10 *Sort*-models still outperforms the baseline using 50 *Sort*-models.

**Storage Efficiency.** Storage Efficiency. Our method (S&S) achieves high storage efficiency, requiring only two 1D arrays: one to store the sort-sum and another to construct the current search output. This results in minimal storage overhead, amounting to just 0.0166% of the input data or less than 100 MB in absolute terms. Consequently, *Sort&Search* not only outperforms alternative methods, such as CopyNearest&Expand, but is also far more memory-optimized.

---

[4]The "compute saved" axis in the plots is computed as $\frac{n}{n'}$. Effective compute savings are: In *Lifelong-CIFAR10*, we do $25,250 \times 1,697,682$ evaluations in the full evaluation v/s $25,250 \times 2,048$ in our evaluation. Similarly, for *Lifelong-ImageNet*, we perform $117 \times 1,986,310$ v/s $117 \times 2,048$ evaluations.

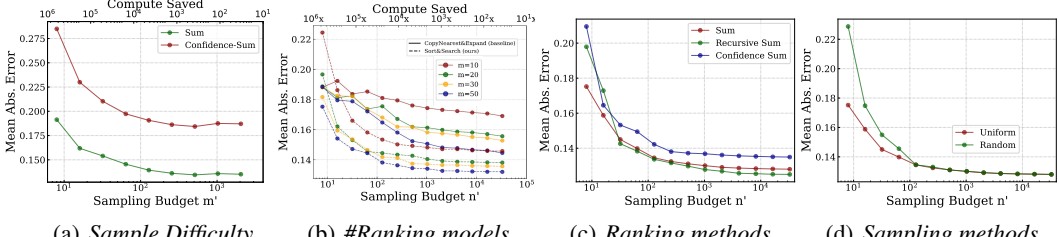

(a) *Sample Difficulty*    (b) *#Ranking models*    (c) *Ranking methods*    (d) *Sampling methods*

Figure 4: *(a)* We achieve accurate sample difficulty estimates on *Lifelong-CIFAR10* ($<$0.15 MAE) at a fraction of the total number of models to be evaluated, thereby enabling cost-efficient sample insertion. *(b,c,d)*, We analyse three design choices for better understanding *S&S*, using *Lifelong-Imagenet*.

## 4.3 Results: Sample Difficulty Estimation (insert$_{\mathcal{D}}$)

We next showcase results for the task (❶) where for new samples, the goal is to sub-sample the number of models to evaluate, for accurately determining sample difficulty. We present results on *Lifelong-CIFAR10*, with two different methods for ranking models[5], *Sorting by Sum* (❶) and *Sorting by Confidence Sum* (❷). We evaluate over 9 model budgets $m'$ (number of models evaluated over): $\{8, 16, 32, 64, 128, 256, 512, 1024, 2048\}$. From Fig. 4(a), we observe that both methods converge quickly—*Sorting by Sum* (❶) reaches an MAE $<$ 0.15 by only evaluating on $m'$=64 models out of $31,250$ ($10^4\times$ computation savings). This demonstrates our method's ability to efficiently determine sample difficulty, enabling efficient insertion back into the lifelong-benchmark pool.

## 4.4 Breaking down *Sort & Search*

**Varying the Number of *Sort*-Models Used.** In Fig. 4(b), we analyse the effect of the number of models used for computing the initial ranking (*i.e.*, $m$) on the final performance on *Lifelong-ImageNet*. Using more models improves MAE— using lesser models for ranking ($m$=10) converges to a higher MAE (2% difference at convergence when using $m$=50 (blue line) vs. $m$=10 (red line)). Note that the $m$ used for ranking does not have any effect on the speed of convergence itself (all methods roughly converge at the same sampling budget ($n'$=2,048)), but rather only on the MAE achieved.

**Different Sorting Methods.** We compare the three different algorithms on *Lifelong-Imagenet*: ❶ *Sorting by Sum*, ❷ *Sorting by Confidence Sum*, and ❸ *Sorting by Recursive Sum*. From Fig. 4(c), we note an MAE degradation when using the continual relaxation of the accuracy prediction values as confidence values, signifying no benefits. However, using the multi-step recursive correction of rankings (❸) provides significant boosts (0.5% boost in MAE at all $n'$$>$1,024) due to its ability to locally correct ranking errors that the global sum method (❶) is unable to account for.

**Different Sampling Methods.** In Fig. 4(d), we compare methods used for sub-selecting the data-samples to evaluate—we compare *uniform* vs. *random* sampling. Both methods converge very quickly and at similar budgets to their optimal values and start plateauing. However, uniform sampling provides large boosts over random sampling when the sampling budget is small (5% lower MAE at $n'$=8)—this can be attributed to its "diversity-seeking" behaviour which helps cover samples from all difficulty ranges, better representing the entire benchmark evaluation samples rather than an unrepresentative random set sampled via random sampling.

## 4.5 Decomposing the Errors of *S&S*

Here, we showcase a decomposition of the errors of *Sort & Search*. Specifically, the total mean absolute error $E(\mathbf{a}_{m+1}, \mathbf{y}_{m+1})$ can be decomposed into a component irreducible by further sampling, referred to as the Aleatoric Sampling Error ($E_{\text{aleatoric}}$), and a component which can be improved by querying larger fraction of samples $n'$, referred to as the Epistemic Sampling Error ($E_{\text{epistemic}}$).

**Aleatoric Sampling Error.** Let $\mathbf{y}^*_{m+1} = \mathbf{y}'$ when $n' = n$, *i.e.*, it is the best prediction obtainable across all subsampled thresholds, as we have access to the full $\mathbf{a}_{m+1}$ vector. However, some error remains between $\mathbf{y}^*$ and $\mathbf{a}_{m+1}$ due to the ordering operation (*i.e.*, *Sort*). This error, caused by errors

---

[5]Recursive sum (❸) is not applicable here as all sum values are unique, see Section 3.3.

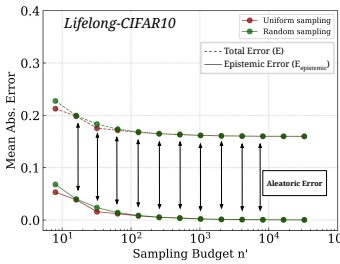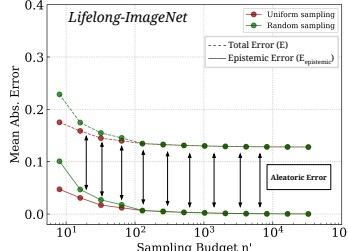

Figure 5: **Error Decomposition Analysis on *Lifelong-CIFAR10* (*left*) and *Lifelong-ImageNet* (*right*).** We observe that epistemic error (solid line) drops to 0 within only 100 to 1000 samples across both datasets, indicating this error cannot be reduced further by better sampling methods. The total error $E$ is almost entirely irreducible (Aleatoric), induced because new models do not perfectly align with the ranking order $\mathbf{P}^*$. This suggests *generalizing beyond a single rank ordering*, ***not** better sampling strategies*, should be the focus of subsequent research efforts.

in the generalization of the permutation matrix $\mathbf{P}^*$ cannot be reduced by increasing the sample budget $n'$. More formally, we define this error as:

$$E_{\text{aleatoric}}(\mathbf{a}_{m+1}, \mathbf{y}_{m+1}) = \min_{\mathbf{y}_{m+1}} \|\mathbf{a}_{m+1}\mathbf{P}^* - \mathbf{y}_{m+1}\| = \|\mathbf{a}_{m+1}\mathbf{P}^* - \mathbf{y}^*_{m+1}\|. \tag{3}$$

**Epistemic Sampling Error.** Contrarily, there is a gap between optimal ranking prediction $\mathbf{y}^*_{m+1}$ and $\mathbf{y}_{m+1}$ with the current sample size $n'$. This gap, Epistemic Sampling Error, is formally defined as:

$$E_{\text{epistemic}}(\mathbf{y}^*_{m+1}, \mathbf{y}_{m+1}) = \|\mathbf{y}^*_{m+1} - \mathbf{y}_{m+1}\|. \tag{4}$$

**Results.** We analyse sampling effectiveness in *Lifelong CIFAR-10* and *Lifelong-ImageNet* by studying the Epistemic Sampling Error ($E_{\text{epistemic}}$) and Aleatoric Sampling Error ($E_{\text{aleatoric}}$) in Figure 5. First, we see that the epistemic error is very low and quickly converges to 0, *i.e.*, we converge to the best achievable performance within sampling just 100 to 1000 samples on both datasets. The remaining error after that is irreducible. We attribute it primarily caused by generalization gaps in the global permutation matrix $\mathbf{P}^*$ as better approximations like *Recursive Sum* (**3**) did not improve performance as shown in Fig. 4(c). This introduces an interesting question: Do models follow a single global ranking order or are they better decomposed into different rank orders?

**How consistently do models follow one single global ranking order?** We present a detailed analysis in Appendix E to verify this. We calculated the cross-correlation matrix for predictions from 167 models across the entire *Lifelong-Imagenet* benchmark (1.9M test samples). Surprisingly, *all model pairs showed positive correlations* to varying degrees, with *no pairs being anti-correlated*. Models with near-zero correlations had near-random performance, indicating uncorrelated predictions due to their randomness. Top-performing models exhibited slightly higher correlations. Overall, there was no clear evidence of model cliques. This analysis strongly suggests that model predictions are highly correlated, justifying our choice of using a single ranking function, but the ranking is simply noisy.

## 5 Conclusion

In this work, we address the efficient lifelong evaluation of models. To mitigate the rising evaluation costs on large-scale benchmarks, we proposed an efficient framework called *Sort & Search*, which leverages previous model predictions to rank and selectively evaluate test samples. Our extensive experiments, involving over 31,000 models, demonstrate that our method reduces evaluation costs by 1000x (over 99.9%) with minimal impact on estimated performance on a sample-level. We aim for *Sort & Search* to inspire the development of more robust and efficient evaluation methods. Our findings show that model predictions are highly correlated, supporting our use of a single ranking function, though the ranking is somewhat noisy. Our analysis of *Sort & Search* suggests that future research should focus on generalizing beyond a single rank ordering, rather than on better sampling strategies. Overall, we hope *Sort & Search* enables large reductions in model evaluation cost and provides promising avenues for future work in lifelong model evaluation.

## Acknowledgements

The authors would like to thank (in alphabetic order): Bruno Andreis, Çağatay Yıldız, Fabio Pizzati, Federico D'Agostino, Ori Press, Shashwat Goel, and Shyamgopal Karthik for helpful feedback. AP is funded by Meta AI Grant No. DFR05540. VU thanks the International Max Planck Research School for Intelligent Systems (IMPRS-IS) and the European Laboratory for Learning and Intelligent Systems (ELLIS) PhD program for support. VU was supported by a Google PhD Fellowship in Machine Intelligence. PT thanks the Royal Academy of Engineering for their support. AB acknowledges the funding from the KAUST Office of Sponsored Research (OSR-CRG2021-4648) and the support from Google Cloud through the Google Gemma 2 Academic Program GCP Credit Award. SA is supported by a Newton Trust Grant. MB acknowledges financial support via the Open Philantropy Foundation funded by the Good Ventures Foundation. This work was supported by the German Research Foundation (DFG): SFB 1233, Robust Vision: Inference Principles and Neural Mechanisms, TP4, project number: 276693517 and the UKRI grant: Turing AI Fellowship EP/W002981/1. MB is a member of the Machine Learning Cluster of Excellence, funded by the Deutsche Forschungsgemeinschaft (DFG, German Research Foundation) under Germany's Excellence Strategy – EXC number 2064/1 – Project number 390727645.

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

# Part I

# Appendix

## Table of Contents

# A  Domain-Agnosticity of Lifelong Benchmarks

Our framework is domain-agnostic. All our framework requires is an $A$ matrix constructed using any binary metric, with rows representing samples and columns representing evaluated models. We discuss several applications of our framework across a range of metrics:

- **Language Models:** Our framework can be directly applied to multiple-choice question evaluations popular for benchmarking language model evaluations. The metric here is exact match or near-exact match, a binary metric that perfectly aligns with our framework requirements.

- **Dense Prediction Tasks or Multi-label Classification:** For pixel-wise prediction tasks or multi-label classification, our framework can be adapted by flattening the predictions of each sample. In this approach, each sample contributes an array of binary values to the $A$ matrix instead of a single value. Extending the search algorithm is straightforward: if a point is sampled, all associated values are sampled and annotated.

- **Tasks with Real-valued Predictions:** For tasks such as regression or BLEU score evaluations, our framework can be used after applying a thresholding operation, which converts predictions into binary values (above or below the threshold). While this adaptation allows the framework to function, it restricts the output predictions to the binary threshold level.

Followup work [3] does extend lifelong benchmarks to evaluating language models and multimodal language models and tackles the unique challenges faced in those cases.

# B   Towards Truly Lifelong Benchmarks: A Conceptual Framework

In the main paper, we introduced the concept of *lifelong model evaluation* through the idea of ever-expanding large-scale benchmarks, termed *Lifelong Benchmarks*. Although *Lifelong-ImageNet* and *Lifelong-CIFAR10* are large-scale, they are not truly lifelong as they do not expand over time. These benchmarks primarily test the efficacy of our *Sort & Search* method due to their large size.

To achieve true lifelong benchmarks, we need continuous acquisition of samples and models, allowing for continual growth (as detailed in Section 2). In Fig. 6, we illustrate how lifelong benchmarking differs from the standard benchmarking approaches currently used in machine learning research.

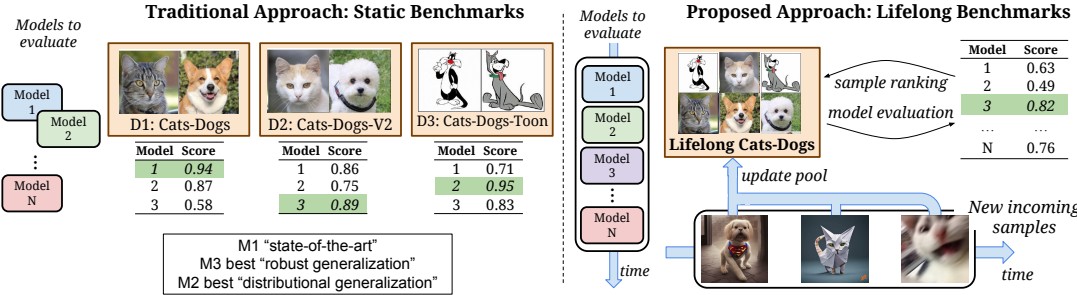

Figure 6: **Static vs Lifelong Benchmarking.** *(Top)* Static benchmarks incentivise machine learning practitioners to overfit models to specific datasets, weakening their ability to assess generalisation. *(Bottom)* We conceptualise *Lifelong Benchmarks* as an alternative paradigm—ever-expanding pools of test samples that resist overfitting while retaining computational tractability.

# C   Lifelong-ImageNet and Lifelong-CIFAR10: Details

In this section, we detail the creation of our two lifelong benchmarks.

**Considerations.** We aim to establish lifelong benchmarking as a standard evaluation protocol in computer vision. To demonstrate this, we considered two popular datasets as our basis: CIFAR10 [54] and ImageNet [23]. We chose them due to (1) their widespread adoption in prior art, (2) the diverse set of models trained on them, and (3) the presence of numerous dataset variants with the same set of labels, encompassing distribution shifts [8], temporal variations [80], and adversarial samples [41].

Note that while our current lifelong benchmarks are based on two datasets, our framework can generally be applied to any broader range of datasets. We describe the precise construction of our datasets below. See Table 1 for key statistics and a detailed breakdown.

**Lifelong-CIFAR10.** We combine 31 domains of different CIFAR10-like datasets comprising samples applied with various synthetic distribution shifts, synthetic samples generated by diffusion models, and samples queried from different search engines using different colors and domains. We deduplicate our dataset to ensure uniqueness and downsample all images to the standard CIFAR10 resolution of $32 \times 32$. Our final dataset consists of 1.69 million samples.

**Lifelong-ImageNet.** We source our test samples from ImageNet and its corresponding variants. Similar to *Lifelong-CIFAR10*, our benchmark is designed for increased sample diversity (43 unique domains) while operating on the same ImageNet class set. We include samples sourced from different web-engines and generated using diffusion models. Our final Lifelong-ImageNet contains 1.98 million samples.

Table 1: **Overview of our Lifelong Benchmarks.** We list the constituent source datasets (deduplicated) and their statistics for constructing our lifelong benchmarks here. Our benchmarks encompass a wide-range of natural and synthetic domains, sources and distribution shifts, making for a comprehensive lifelong testbed.

| Dataset | #Test Samples | #Domains | #Unique Sources | Synthetic/Natural | Corrupted/Clean | License |
|---|---|---|---|---|---|---|
| *Lifelong-CIFAR10* | 1,697,682 | 31 | 9 | Both | Both | |
| CIFAR10.1 [73] | 2,000 | 1 | 1 | Natural | Clean | MIT License |
| CIFAR10 [54] | 10,000 | 1 | 1 | Natural | Clean | Unknown |
| CIFAR10.2 [59] | 12,000 | 1 | 1 | Natural | Clean | Unknown |
| CINIC10 [21] | 210,000 | 1 | 1 | Natural | Clean | MIT License |
| CIFAR10-W [83] | 513,682 | 11 | 8 | Both | Clean | MIT License |
| CIFAR10-C [40] | 950,000 | 19 | 1 | Natural | Corrupted | Apache-2.0 License |
| *Lifelong-ImageNet* | 1,986,310 | 43 | 9 | Both | Both | |
| ImageNet-A [41] | 7,500 | 1 | 3 | Natural | Clean | MIT License |
| ObjectNet [8] | 18,514 | 1 | 1 | Natural | Clean | Custom License |
| OpenImagesNet [55] | 23,104 | 1 | 1 | Natural | Clean | MIT License |
| ImageNet-V2 [74] | 30,000 | 1 | 1 | Natural | Clean | MIT License |
| ImageNet-R [39] | 30,000 | 13 | 1 | Natural | Clean | MIT License |
| ImageNet [23] | 50,000 | 1 | 1 | Natural | Clean | Custom Non-Commercial |
| Greyscale-ImageNet [84] | 50,000 | 1 | 1 | Natural | Clean | MIT License |
| StylizedImageNet [35] | 50,000 | 1 | 1 | Synthetic | Corrupted | MIT License |
| ImageNet-Sketch [96] | 50,889 | 1 | 1 | Natural | Clean | MIT License |
| SDNet [7] | 98,706 | 19 | 1 | Synthetic | Clean | MIT License |
| LaionNet [80] | 677,597 | 1 | 1 | Natural | Clean | Unknown |
| ImageNet-C [38] | 900,000 | 19 | 1 | Natural | Corrupted | Apache-2.0 License |

## D Pythonic Pseudo-code for *Sort & Search* algorithms

Here, we provide pythonic-pseudo code for the constituent algorithms of *Sort & Search*, which we described in detail in Section 3.

```python
def sort_by_sum(A):
    sum_ranking = A.sum(axis=0)
    order = np.flip(np.argsort(sum_ranking))
    return order

def two_stage_sort_by_sum(A, idx):
    #Step 1: Sum
    order = sort_by_sum(A)
    #Step 1: Search
    thresh = dp_search(A[:, order])

    #Iterate over bins
    bins_ordered = sum_bins[order]
    uniq_bins = np.unique(bins_ordered)

    for u_bin in uniq_bins:
        idx = np.nonzero(bins_ordered==u_bin)[0]
        bin_thresh =
        ↪  np.nonzero(np.all([[bins_ordered >=
        ↪  idx.min()],  [bins_ordered <=
        ↪  idx.max()]], axis=0))[1]
        At = A[thresh][:, order[idx]]
        #Step 2: Sum
        new_order = sort_by_sum(At)
        # Replace current ordering within new in
        ↪  bin
        order[idx] = order[idx[new_order]]
    return order
```

```python
def uniform_sampling(query, num_p):
    # idx -> num_p uniformly sampled points
    idx = np.arange(0, len(query),
            len(query)//num_p)[1:]
    return idx

def dp_search(query):
    # query is 1 x k (from a row of PA)
    # (k can be assigned := n, n', m, m')
    query[query==0] = -1
    cumsum = np.cumsum(query)
    idx = np.argmax(cumsum)
    return idx/len(query)
    # threshold as % of length, transfers n' -> n
    ↪  size
```

Listing 1: **(Left)** Algorithm for Optimizing $\mathbf{P}$ given $\mathbf{Y}$ **(Right)** Algorithm for Optimizing $\mathbf{Y}$ given $\mathbf{P}$

# E   Analysis: How Consistently Do Models Follow Global Ranking?

In all our main results using *Sort & Search*, we use a single ranking order for all new models. A natural question arises: *Are all models consistent in their agreement of what is considered a difficult sample, and what is easy?* Perhaps, there could be a clique of models that all agree that certain samples are hard, whereas other models that do not—is this the case or is one ranking order truly sufficient?

To justify this choice of considering a single ranking order, we run a simple experiment. We compute the cross-correlation matrix between each of the 167 models with each other on the predictions across the entire Lifelong-Imagenet benchmark (1,986,310 test samples) where models are sorted in descending order of accuracy i.e. the highest accuracy model is plotted in the first row/column and the least accurate model is plotted last. Note that the 167 models are extremely distinct in architecture, backbone, training datasets, data augmentation, normalization, and loss functions (see full list in Appendix K). The cross-correlation matrix plot is depicted in Fig. 7(b).

**Reading the plot.** The colorbar is important here, it ranges from 0 to 1—we implicitly only look at positively correlated models. We verified that all the correlation values were positive by plotting the distribution of correlation values in Fig. 7(a)—hence, there are no models that are totally anti-correlated with each other. Now, in the correlation matrix, if there exist certain "model cliques"—certain sets of models that are highly correlated with each other and anti-correlated with all others—we would observe disconnected components, systematically isolated squares.

**Result.** From the correlation plot, we do not find any clear evidence of model cliques. The only anomalous entries we could find are low performing models, whose predictions are uncorrelated with all other models as they are random. We observe slightly higher correlations between the top performing models, but note that this is confounded by their high accuracy—if models are highly accurate, their correlations are likely to be higher by chance alone (since there are more ones in the prediction arrays and hence higher chance of intersecting predictions). However, no distinct cliques were found.

Therefore, this analysis further gives us a strong indication that model predictions are highly correlated, hence justifying our choice of using a single ranking function.

**Brief Discussion.** While our analysis suggests that model predictions are highly correlated, we point out that this analysis is done for a varied set of models purely for the task of image classification. We do acknowledge that other tasks like retrieval or captioning might yield different correlation structures, such that there might be different model cliques emerging. Such a structure would then potentially impact our *Sort* algorithm. Hence, while our current results suggest that the sorted order of difficulty generalizes to new incoming models holds fairly robustly, our method might still be sensitive to task deviations, labeling errors etc. We leave a further exploration of this for future work.

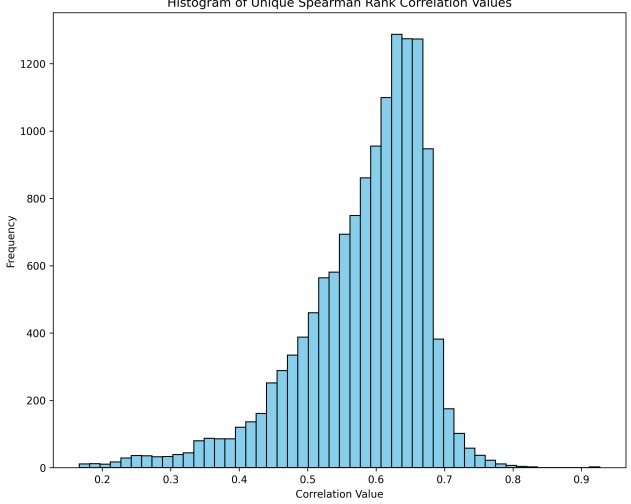

(a) *Spearman Correlations are all positive*

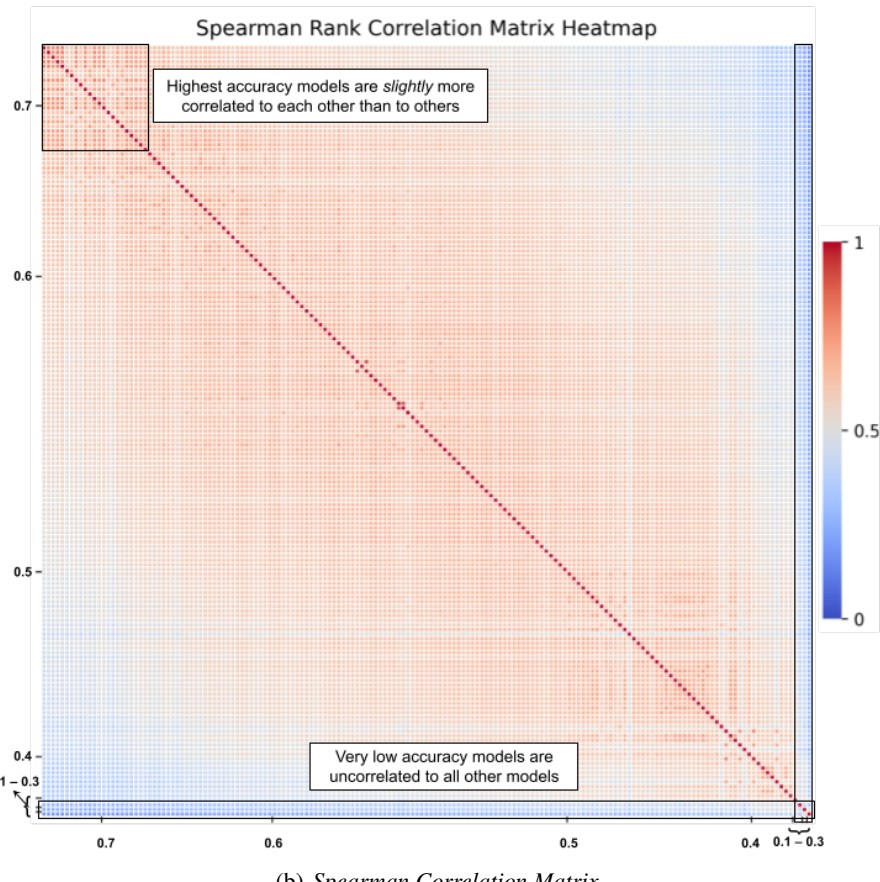

(b) *Spearman Correlation Matrix*

Figure 7: **Correlation Analysis between Model Predictions on *Lifelong-ImageNet*.** *(a)* We note that all correlations between model predictions are positive, signifying the similarities between all models despite their diverse sizes, architectures, and inductive biases. *(b)* We show the cross-correlation matrix between all model predictions—the x and y axes showcase models, sorted by their accuracies. The floating point numbers on the x and y axes are the model accuracies—the highest accuracy models (70% accuracy) appear at the top and left, while the lowest accuracy models appear at the bottom and right (10% − 30%).

# F  Analysis: Changing the metric from MAE to a Rank Correlation

In all our main results using *Sort & Search*, we use the mean-absolute-error (MAE) to evaluate the effectiveness of our framework.

While MAE serves as a useful proxy metric for algorithm development, *it is not a necessary requirement to provide practical applications.* In particular, for many use-cases, it is the *ranking of the models, rather than their absolute metrics, that are of primary importance* for informing downstream decisions about which model to use.

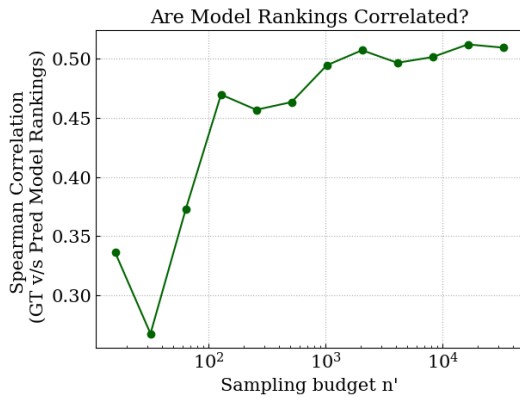

Figure 8: We change the metric for evaluating the efficacy of *Sort & Search* from MAE to Spearman correlation—we observe consistently high correlations of $0.5$ or greater.

To illustrate a practical application, we examine whether *Sort & Search* preserves the ranking of models at high sampling efficiency. Specifically, we conducted an experiment by changing the evaluation metric from MAE to Spearman correlation between the rankings of $25,250$ models using *Sort & Search* and the rankings obtained after full sample evaluation on *Lifelong-CIFAR10*. The results, presented in Fig. 8, show a consistently high correlation of $0.5$. We believe this demonstrates the framework's applicability for practical use-cases.

# G   Does Error Accumulate with Consecutive Additions of New Models/Data?

In this section, we argue that the errors should not accumulate with consecutive addition of new models or data. The core intuition lies in the fact that sequential updates to $\mathbf{P}_t^*$ when made with the predicted vector $\mathbf{y}_{t+1}$ will necessarily preserve the same permutation, i.e. $\mathbf{P}_{t+1}^* = \mathbf{P}_t^*$ as $\mathbf{y}_{t+1}$ strictly follows $\mathbf{P}_t^*$ itself, adding an error of 0.

**Detailed Explanation**. Considering the case where a new model is presented in which $\mathbf{A} \in \{0,1\}^{|\mathcal{M}| \times |\mathcal{D}|}$ where $|\mathcal{M}|$ is the number of models and $|\mathcal{D}|$ the number of data samples. We solve Equation 1 by alternating the solution between solving for $\mathbf{y}$ given the permutation $\mathbf{P}$ and $\mathbf{P}$ given the prediction $\mathbf{y}$. For ease, and without loss of generality, consider the problem when solving Equation 1 repetitively for a sequence of new samples. A natural question is: Do we need to re-optimize for $\mathbf{P}_t$ and update $\mathbf{A}$ with the new ranked prediction vectors $\mathbf{y}_t$ for every timestep?

Our algorithm *Sort & Search*, while might not be achieving global optimality in both $\mathbf{P}$ and $\mathbf{y}$, however, we have a guarantee that if $\mathbf{P}_t^*$ and $\mathbf{y}_t^*$ are the solutions of *Sort & Search* at step t, then $\mathbf{P}_t^* = \mathbf{P}_{t+1}^*$ at every step and we do not require recomputing $\mathbf{P}_{t+1}^*$ optimizing $[\mathbf{A}_t|\mathbf{y}_t^*]\mathbf{P}_{t+1}$ after every addition where $[\mathbf{A}_t|\mathbf{Y}_t^*]$ is the concatenation of $\mathbf{A}_t$ with the new sample $\mathbf{Y}_{t+1}$. This is since *Sort & Search* only requires access to the sum over columns of $[\mathbf{A}_t|\mathbf{Y}_t^*]$ (see Algorithm ❶). The core intuition underlying this result is that at the new step $t+1$ the vector $\mathbf{y}_{t+1}^*$ has a structure of ones followed by zeros ordered according to the optimal permutation $\mathbf{P}_{t+1}^*$ that orders samples from "easiest" to "hardest" following the structure in $\mathbf{AP}_t^*$. Hence, adding it to the sum preserves the ordering of elements (if ties are broken in the manner of the old ordering).

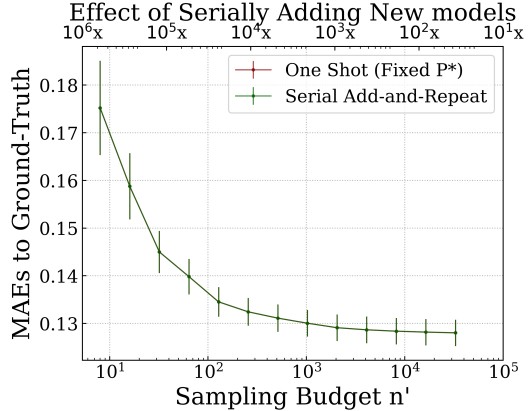

**Empirical Backing.** We conducted experiments by adding new models serially and using the *Sort & Search* predictions as ground truth for further model additions on *Lifelong-ImageNet* dataset. The results are presented in the Appendix G. We observe the errors *do not accumulate* with consecutive additions, exactly the same model order is preserved – confirming our insight empirically.

# H Extended Related Work

In this section, we expand on the brief literature review from Section 2 for a more expansive coverage of related topics.

**Comprehensive Benchmarks.** Benchmarking has become ubiquitous in the machine learning world in the last few years [72]. It has gained further traction in the recent past with the release of foundation models like GPT-4 [18] and CLIP [71]. A popular direction taken by efforts like GLUE [93], BigBench [82], HELM [57] *etc.*, is to have a benchmark of benchmarks, reporting the average accuracy over the constituent datasets. This approach now spans across several domains including fact-based question-answering [40], language understanding [94], zero-shot classification of vision-language models [30], large-scale vision model evaluation [104], multi-modal model evaluation [102, 108], and text-to-image generation [5, 56]. Despite these benchmarks having vast coverage of testing concepts, the obvious downsides are two-fold: (1) they are static in nature and hence can always be susceptible to test-set contamination [60], and (2) their large sizes renders them very expensive to run full model evaluations on.

**Adversarial Dynamic Benchmarks.** One necessary aspect essential for lifelong benchmarks is collecting harder samples, which has been pursued by two strands of works. Adversarial methods to augment benchmarks [92, 63, 51, 70, 81] aim to automatically curate samples that all tested models reliably fail on. These methods usually involve an iterative optimisation procedure to find such adversarial samples. The second strand of work in curating adversarial samples are efforts revolving around red-teaming [31, 66] that aim to explicitly elicit certain sets of behaviours from foundation models; primarily these approaches look at the problem of adversarial benchmarking from a safety perspective. Further, a host of benchmarks that aim to stress-test models are making their way on the horizon—their primary goal is to create test sets for manually discovered failure modes [103, 65, 85, 88, 42, 48, 13, 16]. However, while they are sample efficient, they are criticized as unfair. To mitigate this, a strand of automatic error discovery [19, 27, 99, 68] or their human-in-the-loop variants [97, 24, 32] have been developed. This is complementary to our work, as we primarily explore model testing.

**Active Testing.** Efforts such as [47, 52, 53, 106] aim to identify "high-quality", representative test instances from a large amount of unlabeled data, which can reveal more model failures with less labeling effort. The key assumption underlying these works is that they assume access to a host of unlabeled data at a relatively cheap cost. However, they assume that the cost of label acquisition is a bottleneck. However, these assumptions can break down when doing multiple forward passes on a single batch of data with a large-scale foundation model is necessitated. Albeit similar in spirit to the task of actively acquiring a subset of samples for testing models, an important distinction of our method is that we want to minimise the number of forward-passes through a model—we believe that the cost of running a model on several test samples is substantial, and hence needs to be reduced for efficient evaluation in terms of time, resources and capital.

**Ideas for Replacing Benchmarks.** Recently, there have been a surge of methods introducing creative ways of benchmarking models [58, 76, 49, 33, 75, 77, 61, 44, 17, 86, 64, 34, 76, 75] including hosted competitions [14], self-supervised evaluation [46] and newer metrics [36]. Further, recently ELO style methods have been gaining a lot of attention [12, 107] due to their scalability of deployment to millions of users in a peer-to-peer manner. The ELO algorithm is used to compute ranks for different models based on human-in-the-loop preferences. However, despite its utility ELO is heavily dependent on the choice of user inputs and can be a very biased estimator of model rankings [79]. Another interesting idea proposed by [20] is to assume access to the pre-training data of models and compute topological maps to give predictions of test error; this however requires running expensive forward passes over the training data or modifying the training protocol, which might be not be scalable to pre-trained models.

**Computerized Adaptive Testing.** Computerized Adaptive Testing (CAT) is a framework that allows for efficient testing of human examinees. The idea is to lower the burden of students taking tests by only asking them a subset of questions from the entire pool. There have been few main directions of solutions: model-agnostic strategies for selection [11], bi-level optimization [37, 109, 29], multi-objective optimization [62, 43, 95], retrieval-augmented adaptive search [100]. One key challenge in CAT is the lack of a stable ground-truth. Since the goal in CAT is to estimate the proficiency of an examinee, and the examinee's true ground-truth proficiency is not provided, how would one evaluate the true proficiency of an examinee? Thereby, existing CAT methods cannot explicitly optimise for

predicting ability directly *i.e.* they cannot do exact ability estimation. Hence, CAT methods are not usually guaranteed to converge to the true examinee abilities under certain conditions. The biggest distinction of our work from CAT is the access to the ground-truth targets for the tasks we consider. In both *Lifelong-ImageNet* and *Lifelong-CIFAR10*, we have access to the ground-truth and hence can compute grounded metrics that can be optimised towards, unlike in CAT, where every method has to inherently be label-free.

**Curriculum Learning.** This refers to the problem of finding a curriculum of input samples such that the optimisation objective of an algorithm becomes easier. The most intuitive explanation from curriculum learning comes from how humans learn [50]. In the context of machine learning, the idea behind curriculum learning is to find the "difficulty" of samples, where difficulty is usually defined in terms of the ease of classifying that sample correctly. Some recent works in this direction utilise estimating variance of gradients [1] and other information theoretic properties [26] to estimate sample difficulty. These approaches are complementary to our *Sum* component in *S&S* since these can be easily integrated into our framework directly.

# I  Proof of Theorem 3.1

*Theorem.* **Optimality of Y given P**. For any given $\mathbf{a}_i \in \{0,1\}^{1\times n}$ and $\mathbf{P}$, DP-Search returns an ordered prediction vector $\mathbf{y}_i \in \{0,1\}^{1\times n}$ which is a global minimum of $\|\mathbf{a}_i \mathbf{P} - \mathbf{y}_i\|_1$, where being an ordered prediction vector implies that if $\mathbf{y}_j = 1$ then $\mathbf{y}_{j'} = 1 \forall j' \leq j$. Moreover, if $\mathbf{y}_j = 0$, then $\mathbf{y}_{j'} = 0 \ \forall j' \geq j$.

*Proof.* First, we reduce the problem from Eq. (1) to the following:

$$\mathbf{y}'^* = \operatorname{argmin}_{\mathbf{y}'}\|\mathbf{a}'\mathbf{P}^* - \mathbf{y}'\|$$
$$\text{if} \quad \mathbf{y}'_j = 1, \text{then } \mathbf{y}'_{j'} = 1 \ \forall j' \leq j, \quad \text{and if} \quad \mathbf{y}'_j = 0, \text{then } \mathbf{y}'_{j'} = 0 \ \forall j' \geq j. \tag{5}$$

Note that $\mathbf{y}'$ essentially constructs a vector, $\mathbf{y}'_i$, of all ones up to some index $i$ with the rest being zero . Let $\mathbf{b} = \mathbf{a}'\mathbf{P}^*$ be the sorted vector according to the permutation matrix. Thus, the objective function has the following error:

$$\mathbf{e}(\mathbf{y}'_i) = \left(i - \sum_{k=1}^{i} \mathbf{b}_k\right) + \sum_{k=i+1}^{n} \mathbf{b}_k. \tag{6}$$

Observe that the first term is the number of zeros to the left of index $i$ (inclusive) in $\mathbf{b}$, while the second term is the number of 1s in $\mathbf{b}$ to the right of index $i$.

**Proposition I.1.** *If $\mathbf{y}'_i$ is a minimizer to Theorem 4.2, then, the following holds:*

$$\sum_{k=i+1}^{n} \mathbf{b}_k \leq (n-i) - \sum_{k=i+1}^{n} \mathbf{b}_j.$$

*Proof.* Let $j < i$ and that $\mathbf{y}'_i$ and $\mathbf{y}'_j$ are feasible solutions for Theorem 4.2. However, let that $\mathbf{y}'_i$ be such that the inequality in Proposition I.1 while it is not the case for $\mathbf{y}'_j$. Then, we compare the differences in the objective functions $\mathbf{e}(\mathbf{y}'_i)$ and $\mathbf{e}(\mathbf{y}'_j)$. We have that:

$$\mathbf{e}(\mathbf{y}'_j) - \mathbf{e}(\mathbf{y}'_i) = \left[\left(j - \sum_{k=1}^{j} \mathbf{b}_j\right) + \sum_{k=j+1}^{n} \mathbf{b}_k\right] - \left[\left(i - \sum_{k=1}^{i} \mathbf{b}_k\right) + \sum_{k=i+1}^{n} \mathbf{b}_k\right]$$
$$= 2\sum_{k=j+1}^{i} \mathbf{b}_k - (i-j).$$

However, we know from the assumptions that $2\sum_{i+1}^{n} \mathbf{b}_k \leq n-i$ and that $2\sum_{j+1}^{n} \mathbf{b}_k \geq n-j$. Subtracting the two inequalities we have $2\sum_{k=j+1}^{n} \mathbf{b}_k \geq i-j$ which implies that $\mathbf{y}'(\mathbf{s}_j) \geq \mathbf{e}(\mathbf{y}'_i)$ which implies that $\mathbf{y}'_i$ is a better solution to any other $\mathbf{y}'_j$ not satisfying the inequality in Proposition I.1. $\square$

The inequality condition in proposition I.1 implies that for the choice of index $i$, the number of zeros in $\mathbf{a}$ to the right of index $i$ is more than the number of 1s to the right of index $i$. Since any solution $\mathbf{y}'_i$ either satisfies property in Proposition I.1 or not. Moreover, since Proposition I.1 demonstrated that the set of indices that satisfy this property are better, in objective value, than all those that do not satisfy it, then this condition achieves optimality. $\square$

## J  Proof for Theorem 4.1

*Theorem.* Given any ground-truth vector $\mathbf{a}_{m+1}$, it is possible to construct a prediction vector $\mathbf{y}_{m+1}$ such that $E_{\text{agg}}(\mathbf{y}_{m+1}, \mathbf{a}_{m+1}) = 0$ and $E(\mathbf{a}_{m+1}, \mathbf{y}_{m+1}) = 2\min(1 - |\mathbf{a}_{m+1}|/n, |\mathbf{a}_{m+1}|/n)$

*Proof.* Given $\mathbf{a}_{m+1}$, construct a the prediction vector $\mathbf{y}_{m+1}$, such that $E_{\text{agg}}(\mathbf{y}_{m+1}, \mathbf{a}_{m+1}) = 0$ and $E(\mathbf{a}_{m+1}, \mathbf{y}_{m+1}) = 2.\min(1 - |\mathbf{a}_{m+1}|/n, |\mathbf{a}_{m+1}|/n)$

***Construction:*** We first design construction for the prediction vector $\mathbf{y}_{m+1}$. Let us consider three cases: (i) $|\mathbf{a}_{m+1}| < 0.5$, (ii) $|\mathbf{a}_{m+1}| > 0.5$ and (iii) $|\mathbf{a}_{m+1}| = 0.5$.

*Case 1* ($|\mathbf{a}_{m+1}| < 0.5$): We construct the prediction vector by first flipping all the indexes with value 1 in $\mathbf{a}_{m+1}$ to 0, resulting in MAE of $|\mathbf{a}_{m+1}|/n$. Since, we are constrained to maintain the same $|\mathbf{a}_{m+1}|$, we can flip any $|\mathbf{a}_{m+1}|$ other indexes with values 0 to 1. This is possible in this case as there are more 0s than 1s in $\mathbf{a}_{m+1}$. This results in MAE of $|\mathbf{a}_{m+1}|/n$. Taken together, they achieve the total MAE of $E = 2|\mathbf{a}_{m+1}|/n$.

*Case 2* ($|\mathbf{a}_{m+1}| > 0.5$): We construct the prediction vector by first flipping all the indexes with value 0 in $\mathbf{a}_{m+1}$ to 1, resulting in an MAE of $1 - |\mathbf{a}_{m+1}|/n$. Since, we are constrained to maintain the same $|\mathbf{a}_{m+1}|$, we can flip any other index $1 - |\mathbf{a}_{m+1}|$ with values 1 to 0. This is possible in this case as there are more 1s than 0s in $\mathbf{a}_{m+1}$. This results in an MAE of $1 - |\mathbf{a}_{m+1}|/n$. Taken together, they achieve the total MAE of $E = 2.(1 - |\mathbf{a}_{m+1}|/n)$.

*Case 3* ($|\mathbf{a}_{m+1}| = 0.5$): We construct the prediction vector by flipping all the indexes with value 0 in $\mathbf{a}_{m+1}$ to 1 and flipping all the indexes with value 1 in $\mathbf{a}_{m+1}$ to 0. This achieves the total MAE of $E = 1 = 2|\mathbf{a}_{m+1}|/n = 2.(1 - |\mathbf{a}_{m+1}|/n)$.

This concludes the construction of the prediction vector $\mathbf{y}_{m+1}$.

$\square$

# K  167 Models used for Lifelong-ImageNet experiments

We use the following models (as named in the `timm` [98] and `imagenet-testbed` [84] repositories):

1. BiT-M-R101x3-ILSVRC2012
2. BiT-M-R50x1-ILSVRC2012
3. BiT-M-R50x3-ILSVRC2012
4. FixPNASNet
5. FixResNet50
6. FixResNet50CutMix
7. FixResNet50CutMix_v2
8. FixResNet50_no_adaptation
9. FixResNet50_v2
10. alexnet
11. alexnet_lpf2
12. alexnet_lpf3
13. alexnet_lpf5
14. bninception
15. bninception-imagenet21k
16. cafferesnet101
17. densenet121
18. densenet121_lpf2
19. densenet121_lpf3
20. densenet121_lpf5
21. densenet161
22. densenet169
23. densenet201
24. dpn107
25. dpn131
26. dpn68
27. dpn68b
28. dpn92
29. dpn98
30. efficientnet-b0
31. efficientnet-b0-autoaug
32. efficientnet-b1
33. efficientnet-b1-advprop-autoaug
34. efficientnet-b1-autoaug
35. efficientnet-b2
36. efficientnet-b2-advprop-autoaug
37. efficientnet-b2-autoaug
38. efficientnet-b3
39. efficientnet-b3-advprop-autoaug
40. efficientnet-b3-autoaug
41. efficientnet-b4
42. efficientnet-b4-advprop-autoaug
43. efficientnet-b4-autoaug
44. efficientnet-b5
45. efficientnet-b5-advprop-autoaug
46. efficientnet-b5-autoaug
47. efficientnet-b5-randaug
48. efficientnet-b6-advprop-autoaug
49. efficientnet-b6-autoaug
50. efficientnet-b7-advprop-autoaug
51. efficientnet-b7-autoaug
52. efficientnet-b7-randaug
53. efficientnet-b8-advprop-autoaug
54. fbresnet152
55. inceptionresnetv2
56. inceptionv3
57. inceptionv4
58. instagram-resnext101_32x16d
59. instagram-resnext101_32x32d
60. instagram-resnext101_32x8d
61. mnasnet0_5
62. mnasnet1_0
63. mobilenet_v2
64. mobilenet_v2_lpf3
65. mobilenet_v2_lpf5
66. nasnetalarge
67. nasnetamobile
68. polynet
69. resnet101
70. resnet101_cutmix
71. resnet101_lpf2
72. resnet101_lpf3
73. resnet101_lpf5
74. resnet152
75. resnet18
76. resnet18-rotation-nocrop_40
77. resnet18-rotation-random_30
78. resnet18-rotation-random_40
79. resnet18-rotation-standard_40
80. resnet18-rotation-worst10_30
81. resnet18-rotation-worst10_40
82. resnet18_lpf2
83. resnet18_lpf3
84. resnet18_lpf5
85. resnet18_ssl
86. resnet18_swsl
87. resnet34
88. resnet34_lpf2
89. resnet34_lpf3
90. resnet34_lpf5
91. resnet50
92. resnet50_adv-train-free
93. resnet50_augmix
94. resnet50_aws_baseline
95. resnet50_cutmix
96. resnet50_cutout
97. resnet50_deepaugment
98. resnet50_deepaugment_augmix
99. resnet50_feature_cutmix
100. resnet50_l2_eps3_robust
101. resnet50_linf_eps4_robust
102. resnet50_linf_eps8_robust
103. resnet50_lpf2
104. resnet50_lpf3
105. resnet50_lpf5
106. resnet50_mixup
107. resnet50_ssl
108. resnet50_swsl
109. resnet50_trained_on_SIN
110. resnet50_trained_on_SIN_and_IN
111. resnet50_with_brightness_aws
112. resnet50_with_contrast_aws
113. resnet50_with_defocus_blur_aws
114. resnet50_with_fog_aws
115. resnet50_with_frost_aws
116. resnet50_with_gaussian_noise_aws
117. resnet50_with_greyscale_aws
118. resnet50_with_jpeg_compression_aws
119. resnet50_with_motion_blur_aws
120. resnet50_with_pixelate_aws
121. resnet50_with_saturate_aws
122. resnet50_with_spatter_aws
123. resnet50_with_zoom_blur_aws
124. resnext101_32x16d_ssl
125. resnext101_32x4d
126. resnext101_32x4d_ssl
127. resnext101_32x4d_swsl
128. resnext101_32x8d
129. resnext101_32x8d_ssl
130. resnext101_32x8d_swsl
131. resnext101_64x4d
132. resnext50_32x4d
133. resnext50_32x4d_ssl
134. resnext50_32x4d_swsl
135. se_resnet101
136. se_resnet152
137. se_resnet50
138. se_resnext101_32x4d
139. se_resnext50_32x4d
140. senet154
141. shufflenet_v2_x0_5
142. shufflenet_v2_x1_0
143. squeezenet1_0
144. squeezenet1_1
145. vgg11
146. vgg11_bn
147. vgg13
148. vgg13_bn
149. vgg16
150. vgg16_bn
151. vgg16_bn_lpf2
152. vgg16_bn_lpf3
153. vgg16_bn_lpf5
154. vgg16_lpf2
155. vgg16_lpf3
156. vgg16_lpf5
157. vgg19
158. vgg19_bn
159. wide_resnet101_2
160. xception
161. resnet50_trained_on_SIN_and_IN_then_finetuned_on_IN
162. resnet50_imagenet_subsample_1_of_16_batch64_original_images
163. resnet50_imagenet_subsample_1_of_2_batch64_original_images
164. resnet50_imagenet_subsample_1_of_32_batch64_original_images
165. resnet50_imagenet_subsample_1_of_8_batch64_original_images
166. resnet50_with_gaussian_noise_contrast_motion_blur_jpeg_compression_aws
167. resnet50_imagenet_100percent_batch64_original_images

# L   Limitations and Open Problems

Although showcasing very promising results in enhancing the efficiency of evaluating models on our large-scale Lifelong Benchmarks, our investigation with *S&S* leads to some interesting open problems:

(1) *Ranking Imprecision*: Our error decomposition analysis provides convincing evidence (Section 4.5) that the ordering of samples $\mathbf{P}^*$ while evaluating new models bottlenecks prediction performance. Generalizing from imposing a single sample ordering $\mathbf{P}^*$ to sample ordering structures, such as different clusters of models each with their own orderings or rejection frameworks for models if it does not align with the ordering could dramatically improve the framework.

(2) *Identifying Difficult Samples*: Finding and labeling challenging examples is an essential task for lifelong benchmarks, which is not the focus of our work. Studying hard or adversarial sample selection approaches with lifelong benchmarking is a promising direction. We provide an extensive survey of related approaches in this direction in Appendix H.

(3) *Scaling up to Foundation Models*: Our work mainly tackles lifelong model evaluation under an image classification setting for trained classification models. Despite it being clear that our method should scale to foundation models, since it only relies on the existence of an $A$ matrix, it would be interesting to test it on more benchmarks from the LLM and VLM domain.

