# OpenReview forum: "Efficient Lifelong Model Evaluation in an Era of Rapid Progress"
_NeurIPS.cc/2024/Conference — NeurIPS 2024 poster_

### Official Review · Reviewer_jvk8 · 2024-07-10

**Soundness:** 3
**Presentation:** 2
**Contribution:** 3
**Rating:** 5
**Confidence:** 3

**Summary:**

The paper presents the idea of Lifelong Benchmarks as a way to deal with the problem of model overfitting, both at the individual model level and at the community level. The authors present a framework, Sort & Search (S&S), as a way to deal with the ever increasing benchmarking cost: not only does each benchmark have a myriad of samples on which ML models are tested, but there is an ever increasing number of benchmarks on which to evaluate models, thus exponentially increasing the cost for evaluating model performance. S&S addresses the problems of adding N new samples and M new models to the benchmark, which require evaluating the existing models on the N new samples, and evaluating the M new models with all existing samples, respectively. S&S reduces the complexity of these problems by finding the smallest subsets N' and M' such that the evaluation for the remaining N-N' samples and M-M' models can be extrapolated from the evaluation of N' and M'.

**Strengths:**

**Originality:**

The idea of creating lifelong benchmarks that grow over time as new models and samples become available is interesting and addresses the relevant problem of overfitting. It's a bit unclear, however, where these new samples and models come from and how the datasets actually grow. If a researcher/engineer has to manually update the benchmark/dataset, is it actually that different from simply creating yet another dataset on which to test models?

&NewLine;
**Quality:**

The submission seems to be technically sound and the claims supported. The methods used are appropriate. The authors seem to have conducted quite an extensive evaluation on the proposed approach, however a re-structuring of the evaluation section could be helpful in highlighting all the key results. The authors seem to have struggled with including all evaluation results in the main manuscript, which resulted in a very condensed evaluation section with very little discussion and explanation of the results.

&NewLine;
**Clarity:**

Overall organisation could be improved, particularly in section 3. It is challenging to follow all the reasoning behind the several steps of the framework. It would be really helpful to have a figure highlighting the steps or depicting the framework. Figure 1, for example, helps a lot with understanding the matrices and the model evaluation. Something similar to that is missing for the framework as a whole.

The methodology description of section 3.1 assumes the reader will consult the supplemental material to look at the listings and algorithms, but the main manuscript should be self-contained.

Section 4.2 seems to be out of place, as it describes a design decision of the framework, and not something specific to the experiments.

The baselines and experimental settings could be introduced more concisely at the start of the experimental evaluation section. It would also be nice to have a list/description of the research questions/attributes that are explored in the evaluation section (e.g. RQ1: what is the cost efficiency of S&S, or something like this)

&NewLine;
**Significance:**

Reducing model evaluation cost is a relevant problem that can have significant impact given not only the increasing size of ML models, but also the widespread training and testing of ML models.

**Weaknesses:**

See above for strengths and weaknesses.

**Questions:**

**a)**
Is there an underlying assumption that if a new model M' does well on a "hard" sample s, then it will do well on all samples sorted as easier than s?

**b)**
How do you know/compute matrix Y, the optimal ranked accuracy prediction matrix?

**c)**
Did you come you with DP-Search or did you find it in the literature and applied it to your use-case? I couldn't understand if the algorithm was also a contribution.

**d)**
Could you explain what you mean by "Now, having optimized the ranking, we have a P∗ from the data matrix indicating the sample ordering based on difficulty." (Lines 170-171)

**e)**
Line 191: "We want to find the best n′ observations" -- best with respect to what?

**f)**
You present S&S as a framework. If I wanted to leverage different performance metrics to evaluate the models, how would I go about updating the framework to use the new metric?

&NewLine;
**Comments:**

- [line 131] What does EM stand for?

&NewLine;
**[minor comment / observation]**

- [line 110] Typo on operation 2 -- I believe it should be insert_M

**Limitations:**

The authors could discuss more limitations of the proposed approach. For instance, for what types of tasks does it work? (e.g. only classification or also regression problems?)

---

> ### Author Rebuttal · Authors · 2024-08-07
>
> We sincerely thank the reviewer for their thorough evaluation and positive, enouraging feedback. We seek to address the reviewer’s concerns and questions below:
>
> > S1 It's a bit unclear, however, where these new samples and models come from and how the datasets actually grow. If a researcher/engineer has to manually update the benchmark/dataset, is it actually that different from simply creating yet another dataset on which to test models?
>
> Our inspiration comes from the ChatBot Arena initiative by LMSys, where users test models with inputs from their use cases and provide feedback, which helps grow the benchmark.
>
> Lifelong benchmarks differ from the current paradigm of creating new datasets as they aggregate data from older “solved” benchmarks to compare a wide range of models in a fair manner, rather than discarding old datasets, as new datasets are challenging only for the latest  models.
>
> ---
>
> > S3, point 1: Overall organisation could be improved, particularly in section 3. It is challenging to follow all the reasoning behind the several steps of the framework. It would be really helpful to have a figure highlighting the steps or depicting the framework. Figure 1, for example, helps a lot with understanding the matrices and the model evaluation.
>
> We agree and have added a figure in the common response PDF for reference. We will add this in the revised draft, and believe this will significantly improve the clarity of the sort and search method (section 3). Thank you for highlighting this!
>
> ---
>
> > S3, point 2: The methodology description of section 3.1 assumes the reader will consult the supplemental material to look at the listings and algorithms, but the main manuscript should be self-contained.
>
> We plan to include the framework figure to improve clarity of Section 3, and space permitting we will also include the algorithm in the main paper.
>
> ---
>
> > S3, point 3: Section 4.2 seems to be out of place, as it describes a design decision of the framework, and not something specific to the experiments.
>
> We agree and have shifted this to Section 3. Thank you for pointing this out!
>
> ---
>
> > S3, point 4: The baselines and experimental settings could be introduced more concisely at the start of the experimental evaluation section. It would also be nice to have a list/description of the research questions/attributes that are explored in the evaluation section (e.g. RQ1: what is the cost efficiency of S&S, or something like this)
>
> We will add details about an overview of our results at the start of the experimental evaluation section in the revised draft.
>
> ---
>
> > Q a) Is there an underlying assumption that if a new model M' does well on a "hard" sample s, then it will do well on all samples sorted as easier than s?
>
> Yes, this is the assumption underlying our algorithm. We test the generalizability of permutation matrix $\mathbf{P}^*$ empirically towards new, incoming models in detail in Appendix D: How Consistently Do Models Follow Global Ranking?, where we find that model predictions are correlated, i.e. if a model does well on a “hard” sample s, it does perform well on most samples sorted as easier than s.
>
> ---
>
> > Q b) How do you know/compute matrix Y, the optimal ranked accuracy prediction matrix?
>
> This is explained in detail in l.187 (i) How to get the optimal  ranked accuracy prediction matrix. To summarize: We use DP-search to compute the matrix Y
>
> ---
>
> > Q c) Did you come up with DP-Search or did you find it in the literature and applied it to your use-case? I couldn't understand if the algorithm was also a contribution.
>
> Dynamic programming is a classical set of algorithms (like greedy algorithms). Our contribution is in formulating the solution to the search  as a dynamic programming problem and proving that it is the optimal solution for searching.
>
> On the broader point, our main contributions are as follows:
>
> - Providing Sort&Search, a novel efficient model evaluation on an unexplored setting of lifelong benchmarks
>     - Showing our simple framework is far more scalable and allows saving 1000x evaluation cost.
> - Novel decomposition of errors in Sort&Search into largely independent sub-components (aleatoric and epistemic errors)
>     - Proving theoretically and empirically that our solution for the search subcomponent reaches the optimal solution (in Figure 4).
>
> ---
>
> > Q d) Could you explain what you mean by "Now, having optimized the ranking, we have a P∗ from the data matrix indicating the sample ordering based on difficulty." (Lines 170-171)
>
> To clarify, Section 3.1.2 ended with the theorem. The last statement is a summary of  Section 3.1, which we should clarify by adding a “Summary:” to the start and correcting this to "Now, Ranking by Sort provides a $\mathbf{P}^*$ from the input data matrix, this permutation matrix orders the samples from easy to hard."
>
> ---
>
> > Q e) Line 191: "We want to find the best n′ observations" -- best with respect to what?
>
> We corrected this to “We want to find the $n′$ most informative observations from the $n$ samples” to clarify best and w.r.t what.
>
> ---
>
> > Comment1) [line 131] What does EM stand for?
>
> EM stands for expectation maximization. Sorry for the confusion, we will correct this and write the full term.
>
> ---
>
> > Observation1) [line 110] Typo on operation 2 -- I believe it should be insert_M
>
> Thank you for pointing this out, will correct this right away!
>
> ---
>
> > Limitations 1) The authors could discuss more limitations of the proposed approach. For instance, for what types of tasks does it work? (e.g. only classification or also regression problems?)
>
> Thank you for the question. We discuss extensibility of our setup in comment below-- it is not particularly a limitation of our framework, however present limitations in detail in Appendix I: Limitations & Open Problems.
>
> We hope we have addressed the major concerns of the reviewer, and are happy to answer any further concerns. We look forward to a fruitful reviewer-author discussion phase.

---

> > ### Comment · Reviewer_jvk8 · 2024-08-11
> >
> > Thank you for the clarifications.
> >
> > **Q b)** *"To summarize: We use DP-search to compute the matrix Y"*: Does DP-search give you the global optimum? I mean, if you were to test all models on all samples (and thus knew matrix A unequivocally), would the global optimum be the same as the one computed with DP-search?

---

> ### Author Response · Authors · 2024-08-07
> **Additional Clarification for Q f)**
>
> > Q f) You present S&S as a framework. If I wanted to leverage different performance metrics to evaluate the models, how would I go about updating the framework to use the new metric?
>
> Thank you for raising this important point. All our framework requires is an A matrix constructed using any binary metric, with rows representing samples and columns representing evaluated models. In a sense, this is metric agnostic.
>
> We discuss various applications of our framework which use a wide range of metrics:
>
> - **Language Models:** Our framework can be directly applied to multiple-choice language model evaluations where the metric is exact match or near-exact match, a binary metric perfectly suitable to our framework.
> - **Dense Prediction Tasks or Multi-label Classification:** For pixel-wise prediction tasks or multi-label classification, our framework can be extended by flattening the predictions of each sample. That is, every sample contributes an array of binary values to the A matrix instead of a single value. The extension to the search algorithm is simple, if it samples a point all associated values are sampled and annotated.
> - **Tasks with Real-valued Predictions:** For tasks such as regression or BLEU score evaluations, our framework can operate after applying a thresholding operation. This converts predictions into binary values (above or below the threshold). While this allows the framework to remain valid, it limits the predictions obtained to the binary threshold.
>     - A way to extend this would be having multiple thresholds that can enable quantized searching over output values, but this is beyond the current scope of the work. In contrast, the above applications are more straightforward applications of our framework.
>
> We hope this clarifies the adaptability of our framework.

---

> ### Author Response · Authors · 2024-08-11
> **Reply**
>
> > Q b) "To summarize: We use DP-search to compute the matrix Y": Does DP-search give you the global optimum? I mean, if you were to test all models on all samples (and thus knew matrix A unequivocally), would the global optimum be the same as the one computed with DP-search?
>
> Yes, that is correct. DP-Search returns the optimal $\mathbf{Y}^\*$ for the optimization equation in lines 167-169 if we knew matrix $\mathbf{A}$ unequivocally.
>
> We use precisely this optimal $\mathbf{Y}^\*$ to compute the Aleatoric and Epistemic error discussed in lines 328-332, and results shown in Figure 4. Empirically, we observe that Epistemic error quickly reduces to nearly zero within just 1000 samples.

---

> > ### Comment · Reviewer_jvk8 · 2024-08-13
> >
> > Thank you for the clarifications.

---

### Official Review · Reviewer_tnhy · 2024-07-12

**Soundness:** 3
**Presentation:** 2
**Contribution:** 3
**Rating:** 5
**Confidence:** 3

**Summary:**

To mitigate models from overfitting to the standardized benchmark itself, new samples can be added to the test set, resulting in a Lifelong Benchmark. However, when a new sample is added, all existing models must be evaluated on the added sample. When a new model is added, it must be evaluated on all existing samples. This results in very high evaluation costs. The authors propose a framework termed Sort & search leveraging dynamic programming. First, the samples are ranked w.r.t. their difficulty and sorted accordingly via alternating minimization. When a new model or sample is added, it is only evaluated against a sampled subset and extrapolated, saving computational cost.

**Strengths:**

- The paper discusses about lifelong benchmarking, which is important in mitigating new methodologies and models overfitting to the benchmark itself.
- Computation costs for new evaluations can be effectively reduced (with a trade-off of evaluation error).

**Weaknesses:**

- For Figures 2,3,4, the plots display MAE values larger than $0.1$. In Section 4.6, the paper states that this aleatoric error is irreducible. I am not sure if an MAE of this magnitude is practically tolerable.
- The method assumes that the obtained order of difficulty generalizes to new incoming models (Section 3.2), which might not be the case in the real world. Can any observations be provided on the robustness of the proposed method when this assumption is violated?
- When consecutively adding new models or data serially, is the matrix $\textbf{P}$ and $\textbf{A}$ recomputed after each addition? If so, does the error accumulate for consecutive additions?

**Questions:**

- For Figures 2,3,4, Could the MAE get lower if more computing is utilized? Could the plots be extended further towards $10^{0}$x compute saved (full evaluation)?  (if the experiments are expensive, please do not run them and skip the plot extension part)
- It is relatively difficult to grasp the overall outline of the framework only from the writing, could an outline figure be provided? (if not capable, it is OK)

**Limitations:**

The authors include a limitations section in their manuscript.

---

> ### Author Rebuttal · Authors · 2024-08-07
>
> We sincerely thank the reviewer for their thorough evaluation and their detailed feedback. We seek to address the reviewer’s concerns and suggestions below:
>
> > W1, For Figures 2,3,4, the plots display MAE values larger than 0.1. In Section 4.6, the paper states that this aleatoric error is irreducible. I am not sure if an MAE of this magnitude is practically tolerable.
>
> While MAE serves as a useful proxy metric for algorithm development, it is not a necessary requirement to provide practical applications. In particular, for many use-cases, it is the ranking of the models, rather than their absolute metrics, that are of primary importance for informing downstream decisions about which model to use.
>
> To illustrate a practical application, we examine whether Sort&Search preserves the ranking of models at high sampling efficiency. Specifically, we conducted an experiment by changing the evaluation metric from MAE to Spearman correlation between the rankings of 25,250 models using Sort & Search and the rankings obtained after full sample evaluation on Lifelong-CIFAR10. The results, presented in the attached PDF, show a consistently high correlation of 0.5. We believe this demonstrates the framework's applicability for practical use-cases.
>
> Furthermore, we provide interesting, concrete avenues for improving the sorting algorithm in point (2) in Appendix I – Limitations and Open Problems of our work.
>
> ---
>
> > W2 The method assumes that the obtained order of difficulty generalizes to new incoming models (Section 3.2), which might not be the case in the real world. Can any observations be provided on the robustness of the proposed method when this assumption is violated?
>
> Thank you for raising this point. In Appendix D, "How Consistently Do Models Follow Global Ranking?", we show that the underlying hypothesis of sorted order of difficulty generalizing to new incoming models holds fairly robustly. We will discuss the sensitivity of our sorting algorithms to other factors such as noisy samples and labeling errors, which are important robustness considerations additionally in our revised draft.
>
> ---
>
> > W3 When consecutively adding new models or data serially, is the matrix P and A recomputed after each addition? If so, does the error accumulate for consecutive additions?
>
> Thank you for the great question! We conducted experiments by adding new models serially and using the sort & search predictions as ground truth for further additions. The results are presented in the attached PDF. We observe the *errors do not accumulate with consecutive additions*, exactly the same model order is preserved. We provide an intuitive sketch for why this is the case:
>
> Consider a sum vector $s_t$, which when sorted gives us an order vector $P_t$ at time t. We can prove that sequential updates to the sum array when made with the predicted vector $y_{t+1}$ will necessarily preserve the same $P_t$, i.e. $P_t$ = $P_{t+1}$ for all t.
>
> *Proof Sketch:* The core intuition behind the proof is that vector $y_{t+1}$ is [1111...000] in the order $P_t$, i.e. preserving the order as $P_t$. Incrementing the sum array with a order preserving $y_{t+1}$ preserves the order (if ties are broken in the manner of the old ordering). Why? If an element ${s_i}$ > ${s_j}$ at time t then necessarily ${s_i}$ > ${s_j}$ at t+1 for all elements i,j because y is sorted, i.e. if j >i in y then  ${y_i} >= y_j$.
>
> We shall formalize this and include these results alongside our point (1) in Appendix I – Limitations and Open Problems.
>
> ---
>
> > Q1 For Figures 2,3,4, Could the MAE get lower if more computing is utilized? Could the plots be extended further towards x compute saved (full evaluation)? (if the experiments are expensive, please do not run them and skip the plot extension part)
>
> We have extended Figures 2 and 3 in the common PDF to show the results of a full evaluation. Our observations indicate that the MAE error in these figures cannot be reduced further, demonstrating that *additional sampling does not decrease the MAE*.
>
> ---
>
> > Q2 It is relatively difficult to grasp the overall outline of the framework only from the writing, could an outline figure be provided? (if not capable, it is OK)
>
> We have provided an outline figure in the common PDF. Thank you for pointing this out, we believe this helps clarify the Sort and Search framework.
>
> ---
>
> We hope we have addressed the major concerns of the reviewer, and are happy to answer any further questions/concerns. We look forward to a fruitful reviewer-author discussion phase.

---

> ### Comment · Reviewer_tnhy · 2024-08-10
>
> Thank you for the detailed response. For the author response of W1 and W2,
>
> W1: As the authors have said, MAE serves as a useful proxy metric for algorithm development. I believe that an efficient testing strategy such as the one proposed by the authors would be mainly utilized for quick probing tests during the development phase of an algorithm, for trial & error. For this purpose, approximating the amount of score improvement could be more meaningful than rankings.
>
> W2: My subjective projection on this matter is that for contemporary foundation models such as LLMs, a new version of an LLM could be trained with different proprietary datasets that boost certain abilities of the model. For instance, if a lifelong dataset measures coding abilities, a novel programming language can be introduced, and certain new LLMs may be prepared for this new language, and vice versa.
>
> However, recognizing the Authors' response and the current scope of discourse, I think that it is not necessary to address such aspects of the problem in a single research stride. I will raise my score.

---

### Official Review · Reviewer_T9R1 · 2024-07-12

**Soundness:** 3
**Presentation:** 4
**Contribution:** 3
**Rating:** 7
**Confidence:** 3

**Summary:**

The paper presents a novel approach to addressing overfitting in standardized machine learning benchmarks by introducing Lifelong Benchmarks, which are designed to expand continuously, thereby providing a more dynamic evaluation environment. The authors propose the Sort & Search (S&S) framework for efficient model evaluation on these large-scale benchmarks. The S&S framework reuses previously evaluated models to selectively rank and sub-sample test samples, significantly reducing computational costs. Empirical evaluations on their Lifelong-CIFAR10 and Lifelong-ImageNet benchmarks show that S&S can reduce inference costs with low approximation error. This work contributes to the field by offering a solution for lifelong evaluation, enhancing both model evaluation efficiency and data sample insertion.

**Strengths:**

The paper addresses an important area in the field, offering significant contributions through its novel approach. Specifically, the introduction of the framework called Sort & Search to efficiently evaluate models stands out as a key contribution. It is generally well-written, with well-defined theorems and definitions. The research provides comprehensive experimental evidence to support their claims.

**Weaknesses:**

The paper would benefit from explicitly stating the assumptions about the data samples and models within the main text. I have included specific questions regarding the data samples in the questions section to help clarify these points.

**Questions:**

Q1. Handling Multi-Use/Nested Cases of Benchmarks: The COCO dataset, among others, presents a scenario where each data point is associated with multiple labels, such as categories, super-categories, etc. In the context of lifelong benchmarks, how are these multi-use or nested cases managed? Specifically, would there be multiple versions of COCO to handle each use case separately or would there be just one Lifelong-COCO that handles all annotations? How would these different approaches impact model evaluation and data insertion steps?

Q2. Differentiating Difficult Data Samples from Mismatched Data Samples: When evaluating data samples, how do you distinguish between ‘good difficult’ samples and ‘mismatched’ samples that might be irrelevant or noisy?

**Limitations:**

Yes, the authors adequately addressed the limitations.

---

> ### Author Rebuttal · Authors · 2024-08-07
>
> > W1 The paper would benefit from explicitly stating the assumptions about the data samples and models within the main text. I have included specific questions regarding the data samples in the questions section to help clarify these points.
>
> We thank the reviewer for this point. We have sought to answer the questions raised below for clarity.
>
> > Q1 Handling Multi-Use/Nested Cases of Benchmarks: The COCO dataset, among others, presents a scenario where each data point is associated with multiple labels, such as categories, super-categories, etc. In the context of lifelong benchmarks, how are these multi-use or nested cases managed? Specifically, would there be multiple versions of COCO to handle each use case separately or would there be just one Lifelong-COCO that handles all annotations? How would these different approaches impact model evaluation and data insertion steps?
>
> Thank you for raising this important point. Our framework is domain-agnostic. What we require is an A matrix constructed using any binary metric, with rows representing samples and columns representing evaluated models.
>
> We summarize how our framework can be applied to various domains below:
>
> - **Dense Prediction Tasks or Multi-label Classification:** For pixel-wise prediction tasks or multi-label classification, our framework can be extended by flattening the predictions of each sample. That is, every sample contributes an array of binary values to the A matrix instead of a single value. The extension to the search algorithm is simple, if it samples a point all associated values are sampled and annotated.
> - **Language Models:** Our framework can be directly applied to multiple-choice language model evaluations where the metric is exact match or near-exact match, a binary metric perfectly suitable to our framework.
> - **Tasks with Real-valued Predictions:** For tasks such as regression or BLEU score evaluations, our framework can operate after applying a thresholding operation. This converts predictions into binary values (above or below the threshold). While this allows the framework to remain valid, it limits the predictions obtained to the binary threshold.
>     - A way to extend this would be having multiple thresholds that can enable quantized searching over output values, but this is beyond the current scope of the work. In contrast, the above applications are more straightforward applications of our framework.
>
> We hope this clarifies the adaptability of our framework to various tasks and domains.
>
> ---
>
> > Q2 Differentiating Difficult Data Samples from Mismatched Data Samples: When evaluating data samples, how do you distinguish between ‘good difficult’ samples and ‘mismatched’ samples that might be irrelevant or noisy?
>
> We currently assume that labels are correct, and are unable to identify label noise. One can extend this to noisy samples by trading off sample efficiency, a.la., error-correcting codes. Alternatively, we can apply a cleaning/verification process on the input labels provided by using frameworks like CleanLab and exclude outlier samples for better ranking estimation.
>
> ---
>
> We hope we have addressed the major concerns of the reviewer, and are happy to answer any further questions/concerns. We look forward to a fruitful reviewer-author discussion phase.

---

> > ### Comment · Reviewer_T9R1 · 2024-08-13
> >
> > Thank you for your response.  I echo Reviewer oB2P's opinion and will maintain my current score for the same reasons.

---

### Official Review · Reviewer_bEaJ · 2024-07-18

**Soundness:** 3
**Presentation:** 3
**Contribution:** 2
**Rating:** 5
**Confidence:** 2

**Summary:**

The paper aims to improve the efficiency for evaluating large-scale lifelong benchmarks with the rapidly growing number of machine learning models and samples. To address this issue, the authors propose the Sort & Search framework, which avoids the need to evaluate all samples upon adding new models (or evaluating all models upon inserting new samples). Instead, the proposed Sort & Search framework ranks the samples and selectively applies evaluation to a subset of test samples or models by leveraging previously benchmarked results. The experimental results demonstrate significant improvements in benchmark efficiency, achieving approximately 1000x computational reduction.

**Strengths:**

+ The paper is well-written, with clear examples and theoretical proofs.
+ The paper targets a practical and pressing problem; the experimental results demonstrate significant compute reduction.

**Weaknesses:**

- The evidence for supporting the effectiveness of the Sort algorithm is missing. Specifically, it would be beneficial to compare the proposed method with a sample-only approach to assess if the Mean Squared Error (MSE) will converge to a similar or a larger value. For example, can we skip sorting and directly perform random sampling to get the samples? If “random sampling-only” with a sufficiently large subset (e.g., n’ ~10^3, with moderate compute cost) yields a comparable MSE, the necessity of the sorting algorithm is questionable.
- The reported improvement in MSE is minimal, with only a 0.01 reduction compared to the baseline in Figure 2(c) (e.g., from ~0.14 to 0.13). It is unclear whether such a small improvement justifies the proposed method. What is the reasonable target for MSE, and how challenging is it to achieve a 0.01 improvement?
- The proposed framework saves computational resources at the expense of increased storage overhead, however, the storage overhead is not discussed in the paper.
- The framework appears to be applicable only to classification models, limiting its scope. Can it be extended to other tasks such as object detection or segmentation?
- Minor issue: In Appendix H, the text is overlapped and needs formatting correction.

**Questions:**

1. What are the MSE results of applying uniform/random sampling directly to the sample pool without using the sort algorithm?
2. What is a reasonable target for MSE, and how difficult is it to improve MSE by 0.01?
3. What is the storage overhead introduced by the proposed framework?
4. Can the framework be extended to other tasks such as object detection or segmentation?

**Limitations:**

Yes

---

> ### Author Rebuttal · Authors · 2024-08-07
>
> We sincerely thank the reviewer for their thorough evaluation and positive feedback. We’re pleased that the reviewer recognised our work’s originality, noting it "targets a practical and pressing problem", "experimental results demonstrate significant compute reduction" and "with clear examples and theoretical proofs." We seek to address the reviewer’s concerns and questions below:
>
> > W1/Q1 What are the MSE results of applying uniform/random sampling directly to the sample pool without using the sort algorithm?
>
> We agree with the reviewer that evidence for supporting the effectiveness compared to random is important. We *provide this in our paper*. It is labeled as “CopyNearest&Expand” – we randomly sample n’ points and search using kNN (which replaces the search aspect). We shall revise the naming of this evaluation in the revision to make its role as an important baseline clearer.
>
> We show in Figure 2(c) and 3(b) that our Sort & Search algorithm outperforms this baseline by significant margins, discussed in the next question.
>
> ---
>
> > W2/Q2 What is a reasonable target for MSE, and how difficult is it to improve MSE by 0.01?
>
> Improving the performance by 0.01 MAE requires altering the results on 17,000 samples, which is a large margin. This difficulty is even greater at low MAE levels, as shown in the overall figure PDF.
>
> The Sort and Search method achieved an MAE of 0.14 with 80 samples, while reaching an MAE of 0.13 required 1,000 samples—requiring more than ten times the number of samples for reducing MAE by 0.01.
>
> ---
>
> > W3/Q3 The proposed framework saves computational resources at the expense of increased storage overhead, however, the storage overhead is not discussed in the paper. What is the storage overhead introduced by the proposed framework?
>
> Thank you for this excellent question! Sort & Search only requires storing only two 1D arrays, one which maintains the sort-sum and one array used for constructing the current search output. The storage overhead is hence minimal, being 0.0166% of input data or <100MB in absolute terms. This is indeed a key strength of Sort & Search compared to recent methods including CopyNearest&Expand, and we should emphasize it in our work.
>
> **Details:** Sorting compresses the entire A matrix into a single vector, which can be updated online with a simple sum operation. Searching involves receiving a new vector, selecting n' points, and applying the DP-Search algorithm. These involve storing 3 1D arrays, with additional 1-2 1D arrays required temporarily for evaluation procedures.
>
> ---
>
> > W4/Q4 The framework appears to be applicable only to classification models, limiting its scope. Can it be extended to other tasks such as object detection or segmentation?/Can the framework be extended to other tasks such as object detection or segmentation?
>
> Thank you for raising this important point. Our framework is domain-agnostic. What we require is an A matrix constructed using any binary metric, with rows representing samples and columns representing evaluated models.
>
> We summarize how our framework can be applied to various domains below:
>
> - **Language Models**: Our framework can be directly applied to multiple-choice language model evaluations where the metric is exact match or near-exact match, a binary metric perfectly suitable to our framework.
> - **Dense Prediction Tasks or Multi-label Classification**: For pixel-wise prediction tasks or multi-label classification, our framework can be extended by flattening the predictions of each sample. That is, every sample contributes an array of binary values to the A matrix instead of a single value. The extension to the search algorithm is simple, if it samples a point all associated values are sampled and annotated.
> - **Tasks with Real-valued Predictions**:  For tasks such as regression or BLEU score evaluations, our framework can operate after applying a thresholding operation. This converts predictions into binary values (above or below the threshold). While this allows the framework to remain valid, it limits the predictions obtained to the binary threshold.
>     - A way to extend this would be having multiple thresholds that can enable quantized searching over output values, but this is beyond the current scope of the work. In contrast, the above applications are more straightforward applications of our framework.
>
> We hope this clarifies the adaptability of our framework to various tasks and domains.
>
> ---
>
> > W5 Minor issue: In Appendix H, the text is overlapped and needs formatting correction.
>
> Thank you for bringing this to our notice, we missed this. We have corrected this!
>
> ---
>
> We hope we have addressed the major concerns of the reviewer, and are happy to answer any further questions/concerns. We look forward to a fruitful reviewer-author discussion phase.

---

> > ### Author Response · Authors · 2024-08-12
> > **Gentle Nudge**
> >
> > We would really appreciate it if you could have a look at our replies and let us know if you had any further questions/comments. We  highly value your feedback.

---

### Official Review · Reviewer_oB2P · 2024-07-19

**Soundness:** 4
**Presentation:** 3
**Contribution:** 3
**Rating:** 7
**Confidence:** 3

**Summary:**

The authors propose *lifelong benchmarks*, a solution to the high cost and saturation problems of the current evaluation paradigm. They try to predict which samples will be harder to classify to select a subset that can efficiently serve as a proxy for estimating model performance on the full set, and also use these correlates to determine the importance of new samples for evaluating the existing models.

"Sort & Search" is their proposed method to bring a 1000x reduction in inference cost by finding these representative samples. Each time a new model is added, the "insert" function is called, intended to efficiently find samples to test this new model on to update the cache of sample-level correctness scores, which can be averaged to return a new score for all models in the benchmark.

Using existing information of sample-level difficulty from the performance of initial models, the samples are sorted, using permutation matrix P of the samples that have been compared together across models and prediction matrix Y of which sample/model pairs are correctly/incorrectly scored. By iteratively optimizing Y with constant P and P with constant Y with their DP search algorithm, they can efficiently order the samples by difficulty.

Assuming this ordering will generalize to future models, they can employ uniform or random sampling over the ordering of samples, and they can optimize the selection to pick a set of samples that minimizes the error between the full evaluation over all samples and the smaller set, wrt MAE.

To efficiently insert samples, they just have to evaluate them on a set of models to estimate their difficulty.

**Strengths:**

Strong mathematical formulation and convincing demonstration of the result.

**Weaknesses:**

Simple classification tasks aren't the domain we're most concerned about evaluating models on efficiently. It is unclear how this would extend to harder domains such as LM evaluation.

**Questions:**

None

**Limitations:**

Limitations addressed my weaknesses.

---

> ### Author Rebuttal · Authors · 2024-08-07
>
> We thank the reviewer for their detailed feedback. We seek to address the reviewer’s concerns and suggestions below:
>
> > W1 Simple classification tasks aren't the domain we're most concerned about evaluating models on efficiently. It is unclear how this would extend to harder domains such as LM evaluation.
>
> Thank you for raising this important point. Our framework is domain-agnostic. What we require is an A matrix constructed using any binary metric, with rows representing samples and columns representing evaluated models. We summarize how our framework can be applied to various domains below:
>
> - **Language Models**: Our framework can be directly applied to multiple-choice language model evaluations where the metric is exact match or near-exact match, a binary metric perfectly suitable to our framework.
> - **Dense Prediction Tasks or Multi-label Classification**: For pixel-wise prediction tasks or multi-label classification, our framework can be extended by flattening the predictions of each sample. That is, every sample contributes an array of binary values to the A matrix instead of a single value. The extension to the search algorithm is simple, if it samples a point all associated values are sampled and annotated.
> - **Tasks with Real-valued Predictions**:  For tasks such as regression or BLEU score evaluations, our framework can operate after applying a thresholding operation. This converts predictions into binary values (above or below the threshold). While this allows the framework to remain valid, it limits the predictions obtained to the binary threshold.
>     - A way to extend this would be having multiple thresholds that can enable quantized searching over output values, but this is beyond the current scope of the work. In contrast, the above applications are more straightforward applications of our framework.
> We hope this clarifies the adaptability of our framework to various tasks and domains, including language modeling.
>
> ---
>
> We hope we addressed the major concerns of the reviewer, and are happy to answer any further questions/concerns. Looking forward to a fruitful reviewer-author discussion phase.

---

> > ### Comment · Reviewer_oB2P · 2024-08-13
> >
> > Thank you for your response. Yes, I agree that it is easy to see how this technique could be applied to those consequential problems; that's why I gave a pretty high score to begin with. While I think the paper would have been much more impactful if the technique were demonstrated on these tasks, I think the current version as is already is a solid paper that should be accepted. I will keep my score.

---

### Author Rebuttal · Authors · 2024-08-07

We thank the reviewers for finding our work to be ***important in the era of large models*** (Reviewers jvk8, T9R1, bEaJ, tnhy), to have ***strong mathematical formulations and theoretical results*** (Reviewers oB2P, bEaJ, T9R1), and to be ***tackling an important pressing problem with sound empirical results*** (Reviewers jvk8, T9R1, bEaJ). We provide detailed responses to each individual reviewer independently, and summarize the most important common points and additional experiments and visualizations provided in the rebuttal here:

- We have added a **detailed visualization of the Sort & Search method** in the uploaded PDF. We hope that this figure improves the clarity of our paper, and we will ensure to add this into section 3 of the revised version.
- We emphasise that **our framework is domain- and task-agnostic**. We summarize below how our Sort & Search framework can be extended to tasks beyond classification.
  - **Language Models**: Our framework can be directly applied to multiple-choice language model evaluations where the metric is exact match or near-exact match, a binary metric perfectly suitable to our framework.
  - **Dense Prediction Tasks or Multi-label Classification**: For pixel-wise prediction tasks or multi-label classification, our framework can be extended by flattening the predictions of each sample. That is, every sample contributes an array of binary values to the A matrix instead of a single value. The extension to the search algorithm is simple, if it samples a point all associated values are sampled and annotated.
  - **Tasks with Real-valued Predictions**: For tasks such as regression or BLEU score evaluations, our framework can operate after applying a thresholding operation. This converts predictions into binary values (above or below the threshold). While this allows the framework to remain valid, it limits the predictions obtained to the binary threshold. A way to extend this would be having multiple thresholds that can enable quantized searching over output values, but this is beyond the current scope of the work. In contrast, the above applications are more straightforward applications of our framework. We hope this clarifies the adaptability of our framework to various tasks and domains, including language modeling.
- We extend our results from fig 2 to n’={64,000, 128,000, 256,000, 512,000} and add these plots in the common PDF. We observe that the **absolute error (MAE) does not further reduce as we increase the sampling budget**, corroborating the point in sec 4.6 that **additional sampling does not decrease the MAE**.
- We conduct an additional experiment to **serially add new models using the Sort & Search** predictions as ground truth for further additions. We observe that the **errors do not accumulate with consecutive serial applications of our Sort & Search framework**. We provide a simple intuitive proof for why this is the case. This further **showcases the robustness of our method in being applicable without introducing cascading errors**.

---

### Decision · Program_Chairs · 2024-09-25

**Decision:**

Accept (poster)

**Comment:**

The authors propose an efficient evaluation framework, Sort & Search (S&S), to reduce the computational cost of evaluating a growing number of models on an ever-expanding sample set. The framework leverages dynamic programming to selectively rank and sub-select test samples, enabling efficient lifelong benchmarking.

Given the overall positive reviews, I recommend accepting the paper.